



# Global sea level reconstruction for 1900-2015 reveals regional variability in ocean dynamics and an unprecedented long weakening in the Gulf Stream flow since the 1990s

Tal Ezer[1], Sonke Dangendorf[1]

[1]Center for Coastal Physical Oceanography, Old Dominion University
4111 Monarch Way, Norfolk, Virginia, 23508, USA

Corresponding author: Tal Ezer (tezer@odu.edu)

Submitted to *Ocean Science* on March 23, 2020



**Abstract**. A new monthly global sea level reconstruction for 1900-2015 was analyzed and compared with various observations to examine regional variability and trends in the ocean dynamics of the western North Atlantic Ocean and the U.S. East Coast. A proxy of the Gulf Stream (GS) strength in the Mid-Atlantic Bight (GS-MAB) and in the South Atlantic Bight (GS-SAB) were derived from sea level differences across the GS in the two regions. While decadal oscillations dominate the 116-year record, the analysis showed an unprecedented long period of weakening in the GS flow since the late 1990s. The only other period of long weakening in the record was during the 1960s-1970s. Ensemble Empirical Mode Decomposition (EEMD) was used to separate oscillations at different time scales, showing that the low-frequency variability of the GS is connected to the Atlantic Multidecadal Oscillations (AMO) and the Atlantic Meridional Overturning Circulation (AMOC). The recent weakening of the reconstructed GS-MAB was mostly influenced by weakening of the upper mid-ocean transport component of AMOC as observed by the RAPID measurements for 2005-2015. Comparison between the reconstructed sea level near the coast and tide gauge data for 1927-2015 showed that the reconstruction underestimated observed coastal sea level variability for time scales less than ~5 years, but lower frequency variability of coastal sea level was captured very well in both amplitude and phase by the reconstruction. Comparison between the GS-SAB proxy and the observed Florida Current transport for 1982-2015 also showed significant correlations for oscillations with periods longer than ~5 years. The study demonstrated that despite the coarse horizontal resolution of the global reconstruction (1°x1°), long-term variations in regional dynamics can be captured quite well, thus making the data useful for studies of long-term variability in other regions as well.

## 1. Introduction

Various analyses of tide gauge data show acceleration is global sea level rise over the past century with especially significant acceleration in recent years (Church and White, 2006, 2011; Merrifield et al., 2009; Jevrejeva et al., 2008; Woodworth et al., 2011; Hay et al., 2015; Dangendorf et al., 2019). However, the presence of pronounced natural variability at various timescales makes the detection of the long-term acceleration due to anthropogenic climate change more difficult with existing sea level data (Kopp, 2013; Dangendorf et al., 2014; Haigh et al., 2014; Kenigson and Han, 2014). Evaluating global sea level acceleration is important for understanding the global climate system but knowing the mean global sea level rise is insufficient for preparation of coastal communities under threat of increased





flooding. Other factors such as land subsidence and ocean and atmospheric dynamics can have
significant impact on regional relative sea level rise, introducing substantial differences to global sea
level rise (Cazenave and Cozannet, 2014).

13        The U.S. East Coast is a region that has been recently labeled as a "hotspot for accelerated sea

level rise" (Boon, 2012; Ezer and Corlett, 2012; Sallenger et al., 2012; Kopp, 2013; Ezer, 2013; Ezer et
al., 2013; Gehrels et al., 2020). Land subsidence associated with the Glacial Isostatic Adjustment (GIA)
plus local geological, cryospheric and hydrological processes increase local sea level rise along the U.S.
East Coast relative to the global rates (Boon et al., 2010; Kopp, 2013; Miller et al., 2013; Frederikse et
al., 2017; Gehrels et al., 2020). An additional factor, less understood, is acceleration/deceleration due to
the dynamic response to changes in ocean circulation, for example, a potential slowdown in the GS and
AMOC can increase coastal sea level along the western North Atlantic coasts (Ezer and Corlett, 2012;
Sallenger et al., 2012; Ezer et al., 2013; Ezer and Atkinson, 2014; Rahmstorf et al., 2015; Little et al.,
2019). Therefore, it is important to study regional climatic changes for flood-prone coastal communities.
The idea of connections between weakening in the GS strength and rising coastal sea level is not new
(Blaha, 1984) and has been identified in data and ocean models (Ezer, 1999, 2001, 2013, 2015; Ezer at
al., 2013; Levermann et al., 2005; Yin et al., 2009; Yin and Goddard, 2013; Griffies et al., 2014;
Goddard et al., 2015). Because sea level is lower/higher on the onshore/offshore side of the GS (by ~1-
1.5 m; due to the geostrophic balance), changes in the path and strength of the GS are expected to
impact coastal sea level variations along the U.S. East coast. This connection involves various temporal
and spatial scales and complex mechanisms, so detecting the exact fingerprints of changes in the AMOC
and the GS is still an ongoing research (e.g., Little et al., 2019; Piecuch et al., 2019). The processes that
transfer large-scale open-ocean signals into coherent regional coastal sea level response involve short-
term barotropic deep ocean waves, long-term baroclinic waves and coastally trapped waves (Huthnance,
1978; Ezer, 2016; Hughes et al., 2019). Variations in the GS flow and path have a wide range of time
scales: daily, mesoscale, seasonal, interannual, decadal and multidecadal. However, since direct
continuous observations of the GS are relatively short, about 3 decades of satellite altimeter data and
about 4 decades of cable observations of the Florida Current (Baringer and Larsen, 2001; Meinen et al.,
2010), it is difficult to study past decadal and multidecadal variability in ocean dynamics and compare it
to current and future climate change. For example, limited past temperature and salinity ship
observations and simple diagnostic numerical ocean models suggested that a dramatic decline of ~30%
in the GS transport happened between the 1960s and 1970s (Levitus, 1989, 1990; Greatbatch et al.,



1991); at the same period, an increase in sea level along the U.S. east coast of 5-10 cm was observed
(Ezer et al., 1995). These changes resemble recent changes, but observations of the GS and AMOC were
not available at the time, to allow comparisons with recent changes. Using ocean models forced by
surface observations since the 1960s Blaker et al. (2014) found similarities between the extreme minima
in AMOC in 2009/2010 and a similar minima in 1969/1970, but this approach has some shortcomings
due to models' errors and lack of accurate surface forcing for earlier years.

47        One approach to overcome the above limitations of studying long term past changes, is to take

advantage of the global coverage of recent altimeter data and combine this data with sparse, but long,
tide gauges, to obtain global sea level reconstructions. Various optimization and spatial analysis
methods were used to produce global reconstructed sea level (Church et al., 2011; Calafat et al., 2014;
Hamlington et al., 2014; Hay et al., 2015; Dangendorf et al., 2019). Here, we used the latest hybrid
reconstruction of Dangendorf et al. (2019) (see more details in the next section), since it contains both,
spatial and temporal variability, as well as long term trends in sea level. Note that this monthly global
reconstruction excludes non-climatic land motion, excludes seasonal cycles and is currently available at
1°x1° resolution for 1900-2015 (future improvements with higher resolution and an extended period are
planned). Dangendorf et al. (2019) used this reconstruction to study global sea level acceleration and the
influence of southern hemisphere winds on sea level. The main goal here is to evaluate the usefulness of
this reconstruction to study processes of long-term regional ocean dynamics. The southwestern North
Atlantic Ocean was chosen as a test case because of the important role that the GS and AMOC play and
the fact that the nearby U.S. coasts are considered "hotspots" for sea level rise, as described above.
Some questions that the study addresses include: 1. Can a coarse resolution reconstruction that does not
resolve sharp fronts like that of the GS be able to capture dynamic variations in a western boundary
current? 2. How well does the reconstruction, which rely only on altimeter data and sparse tide gauge
data, compared with recent independent observations of Atlantic Ocean circulation features such as the
AMOC and the Florida Current? 3. What characterizes the long-term variability of sea level and ocean
dynamics and how do recent changes such as weakening AMOC compare with past changes? (are recent
changes unprecedented, or more likely natural modes comparable to past changes over the last
century?).

69        The paper is organized as follows: first, the data and the analysis methods are described in

section 2, then in section 3, results are presented for analysis of the entire 116 years record and for



comparisons with observations of recent decades, and finally in section 4, summery and conclusions are
offered.

**2. Data sources and analysis methods**
The global reconstructed sea level (RecSL) record (1900-2015) analyzed here is described by
Dangendorf et al. (2019). This RecSL is an improved hybrid reconstruction based on 479 tide gauge
records, satellite altimeter data, and several geophysical ancillary datasets of contributing processes (e.g.
gravitational, rotational, and deformational effects of mass changes known as "fingerprints", ocean
circulation models and GIA), combining the techniques of the Kalman Smoother (Hay et al., 2015),
optimal interpolation and empirical orthogonal functions (Calafat et al., 2014) at different timescales.
The result is a monthly sea level field on a (1°x1°) grid that includes both, variability and trend (though
the annual cycle was removed). The aim here is to examine this global data set for its usefulness in
studies of regional ocean dynamics. The western North Atlantic region is characterized by strong
mesoscale variability, an intense western boundary current (the Gulf Stream) and important coastal
impacts from climate change and sea level rise along the U.S. East Coast. Therefore, it is a challenging
task for a coarse resolution reconstruction, which does not directly resolve mesoscale features, to
accurately represent the regional dynamics (indirectly though, the tide gauge data may include
contributions from mesoscale dynamics).
From the reconstructed sea level, a proxy of the GS strength was derived for two regions. Based
on the assumption that the surface flow is close to geostrophic balance, the sea level gradient across the
GS represents the strength of the surface GS. In the Mid-Atlantic Bight (MAB), for each longitude the
GS location is defined by the maximum north-south sea level gradient, so the averaged maximum
gradient represents the mean eastward flowing GS in the region (58°W-70°W, 36°N-40°N). The units
are change in cm per 1° latitude. In the South-Atlantic Bight (SAB) similar latitudinal averaging of east-
west gradients will represent the mean northward flowing GS in the region (76°W-80°W, 28°N-32°N),
i.e., between the Florida Strait and Cape Hatteras. These two proxies will be referred to as GS-MAB and
GS-SAB, respectively.
The monthly mean sea-level record (1927-2015) for the tide gauge station in Norfolk
(76.33°W, 36.95°N) was obtained from the Permanent Service for Mean Sea-level (PSMSL,
www.psmsl.org; Woodworth and Player, 2003). Seasonal variations were removed from the data, so



they can be compared with the RecSL record. The Norfolk station at the southern end of the Chesapeake Bay was chosen because it is one of the locations with large acceleration in flooding and one of the U.S. cities currently facing some of the largest impacts of sea level rise. The record was subject to numerous studies that link coastal sea level there with changes in ocean dynamics (Ezer, 2001, 2013; Ezer and Corlett, 2012; Ezer et al., 2013; Ezer and Atkinson, 2014).

The Atlantic Meridional Overturning Circulation (AMOC) data was obtained from the RAPID observations at 26.5°N for 2005-2015, as described in various studies (https://www.rapid.ac.uk/; McCarthy et al., 2012; Srokosz et al., 2012; Smeed et al., 2014). The AMOC transport (given in Sverdrup; 1 Sv = $10^6$ m$^3$ s$^{-1}$) is the sum of three components: 1. The upper mid-ocean transport obtained from observations of density changes across the Atlantic Ocean, 2. the Ekman transport estimated from wind stress data, and 3. The Gulf Stream transport obtained from cable measurements of the Florida Current across the Florida Strait. The twice-daily data of the three components and the total were used to calculate monthly averages.

The monthly Atlantic Multi-decadal Oscillation (AMO) index (Enfield et al., 2001) for 1900-2015 was obtained from NOAA (https://www.esrl.noaa.gov/psd/data/timeseries/AMO/); AMO represents variations in the sea surface temperature (SST) over the Atlantic Ocean. Long-term variations in sea level, such as the ~60-year long cycle, are thought of being influenced by AMO (Chambers et al., 2012) and correlations of AMO with patterns of sea level along the U.S. and European coasts are often indicated (Ezer et al., 2016; Han et al., 2019).

Daily observations of the Florida Current (FC) transport at ~27°N for 1982-2015 were obtained from NOAA/AOML (www.aoml.noaa.gov/phod/floridacurrent/); the data is described by Baringer and Larsen (2000), Meinen et al. (2010) and many other studies. Monthly averaged values were calculated to allow comparisons with the RecSL record. Note that the FC data has a gap from October 1998 to June 2000, but since our focus was on decadal and longer variations the gap did not pose a significant problem in the analysis.

A useful tool to analyze non-linear time series is the Empirical Mode Decomposition (EMD (Huang et al., 1998; Wu et al., 2007), where a repeated sifting process decomposes records into a finite number of intrinsic oscillatory modes $c_i(t)$ and a residual "trend" $r(t)$. The number of modes depends on the record length and the variability of the data. Unlike regression fitting methods, the shape of the trend is not predetermined (i.e., the method is "non-parametric"). Each individual mode does not necessarily


represent a particular physical process, but often a group of modes can be shown to relate to a known
forcing (Ezer et al., 2013; Ezer, 2015). The EMD decomposes the original time series into modes
$$\eta(t) = \sum_{i=2}^{N-1} c_i(t) + r(t).$$ (1)
In the EMD analysis output, mode-1 will be the original time series ($\eta$), modes 2 to N-1 are oscillating
modes with different frequencies from high to low and mode-N will be the trend (r). Combining several
low-frequency modes will be equivalent to a low-pass filter. Note that unlike spectral analysis, the
frequency and amplitude in each mode is not constant, thus the analysis can capture non-linear changes,
such as climatic changes in the amplitude of decadal variability. An improved version of the original
EMD, is the Ensemble EMD (EEMD; Wu and Huang, 2009) used here, where ensemble of simulations
with white noise are averaged. Here, 100 ensemble members are used with white noise of 0.1 of the
standard deviation (see Ezer and Corlett, 2012 and Ezer et al., 2016, for sensitivity experiments with
EEMD parameters and error estimations). The EEMD filters out unphysical modes and is more accurate
for detecting real low frequency variability (Kenigson and Han, 2014). All the calculations here use the
EEMD, though for simplicity the text refers to "EMD".

**3. Results**
**3.1. Sea level rise and Gulf Stream variability 1900-2015**
Using the same reconstruction (RecSL) analyzed here, Dangendorf et al. (2019) found besides
substantial decadal variability a significant and persistent acceleration in global mean sea level since the
1960s. They attributed the initiation of this recent acceleration to shifts in Southern Hemispheric wind
patterns driving changes in ocean circulation increasing the ocean's heat uptake. In the western North
Atlantic, some studies suggest that acceleration in sea level along the eastern coasts of North America
may be related to a slowdown of AMOC and the GS (Leverman et al., 2005; Boon, 2012; Ezer and
Corlett, 2012; Sallenger et al., 2012; Yin et al., 2013; Caesar et al., 2018). Because future projections
from climate models consistently indicate a weakening AMOC (Cheng et al., 2013; Reintges et al.,
2017), it is important to understand the AMOC-sea level connection and try to detect current and past
changes from observations. To evaluate regional patterns in sea level rise, the sea level change in the
southwestern North Atlantic for different periods was calculated (Fig. 1a-e) as well as the sea level



change for the entire record 1900-2015 (Fig. 1f). Two findings emerge from this analysis: First, sea level
is rising at very different rates during different periods, for example, from 1915 to 1935 (Fig. 1a) sea
level rose in the southwestern North Atlantic region by ~0.02-0.04 m (rate of ~1-2 mm/y; similar to the
global rate seen in Fig. 2 of Dangendorf et al., 2019), while from 1995 to 2015 (Fig. 1e) sea level in this
region rose by ~0.05-0.2 m (rate of 2.5-10 mm/y). Therefore, there is clear acceleration of sea level over
the entire period, but this acceleration is spatially very uneven (Fig. 1f). It also seems that due to decadal
variability, some periods experienced even deceleration, for example, sea level rise from 1955 to 1975
(Fig. 1c) was slower than sea level rise from 1935-1955 (Fig. 1b). Second, the largest changes are seen
near the GS around 35°N-40°N. The total sea level change between the first and last 5 years of the RSL
record (Fig. 1f) shows a sea level rise north of the GS and a sea level drop south of the GS, thus
indicating a weakening trend in the geostrophic surface flow of the GS.

A comparison of the global monthly mean sea level with the regional mean sea level in the
southwestern North Atlantic (the area shown in Fig. 1) indicates a similar general trend (Fig. 2a), but
much larger interannual and decadal variability of up to ±4 cm over the global mean sea level (Fig. 2b).
Regionally lower than average sea level is seen in the 1920s-1940s and higher than average sea level in
the 1960s-1980s. Low-passed filtered data (using EMD modes) shows variations in two major period-
bands of ~5-10 years and 10-60 years. The decadal and multidecadal variations in the global
acceleration/deceleration of sea level were described by Dangendorf et al. (2019) and others, but we
further want to evaluate here if regional variations in ocean dynamics may play a role and how these
variations are connected to basin-scale climate modes (Han et al., 2019).

Variability in the GS strength in the MAB (a proxy obtained from sea level gradients, as
described in section 2) is shown in Fig. 3a, indicating large variability on interannual and decadal time
scales with a persistent weakening trend since ~1990, after a period of strengthening flow from the
1970s to the 1990s. The changes in the low-frequency oscillations are shown in Fig. 3b, indicating two
long periods with declining GS strength (red area) during the 1960s and 1970s and after ~1995, with
maximum weakening of ~25% per decade. Recent observations by Andres et al. (2020) at 68.5°W found
the GS transport to be about 10% weaker today than it was in the 1980s at the same location, but the
same study also found very large discrepancy in the trend between two sections located just a few 100
km from each other, a western section from ship crossing showed no statistically significant trend
(Rossby et al., 2014) and an eastern section from mooring data showed potential weakening of ~5-10%
per decade. Based on altimeter data, Dong et al. (2019) and Zhang et al. (2020) also showed different





trends between the eastern and western parts of the GS. Therefore, average GS proxy over a large area
as done here may filter out spatial variations; the RecSL record is also much longer than the altimeter
data used in the above studies. The course resolution of the reconstruction also served as a filter that
smoothed out small spatial variations and impact from local recirculation gyres as seen in Andres et al.
(2020). The GS proxy here shows that the recent weakening period is the longest in this record; it is
generally consistent with other studies that show recent weakening in the GS flow, the subpolar gyre
circulation and AMOC transport (Hakkinen and Rhines, 2004; Bryden, 2005; McCarthy et al., 2012;
Srokosz et al., 2012; Ezer et al., 2013; Smeed et al., 2014; Blaker et al., 2014; Roberts et al., 2014; Ezer,
2015; Dong et al., 2019; Rahmstorf et al., 2015; Caesar et al., 2018). The earlier period of GS
weakening in the 1960s-1970s is consistent with observations and models that showed large density
changes in the North Atlantic and as much as 30% weakening in the GS between 1955-1959 and 1970-
1974 (Levitus, 1989, 1990; Greatbatch et al., 1991; Ezer et al., 1995). At the time of these early studies,
before the age of satellite altimeters, observations were limited and models less sophisticated, so there
were some doubts that the large weakening in the GS during the 1960s and 1970s was real. However,
this reconstruction by Dangendorf et al. (2019) and another reconstruction of AMOC from sea level data
by Ezer (2015) both confirm the results of the early studies, showing only two periods of significant
weakening AMOC since the 1950s.

207       The large decadal and multidecadal variations in the GS proxy are compared with the monthly

Atlantic Multi-decadal Oscillation index (AMO; Enfield et al., 2001) for 1900-2015 (Fig. 4). EEMD is
used to compare oscillating modes with similar time scales. Hi-frequency modes of the GS and AMO
are not significantly correlated, but variability in the two time series on time scales of ~10-60 years are
correlated, especially the lowest frequency modes (bottom two panels in Fig. 4), with correlations of
0.5-0.8 that are statistically significant (after considering the reduction in degrees of freedom in the low-
frequency modes). Mode 6 (bottom panel in Fig. 4) indicates cyclic behavior at periods up to ~60 years,
consistent with previous studies (Chambers et al., 2012). Various studies indicated connection between
AMO, which represents variations in SST, and sea level. Ezer et al. (2016) for example, showed a
change in the sign of the correlation across the GS, which could indicate changes in the GS strength; if
sea level rises at one side of the GS and drops at the other side, the change in gradient indicates a change
in strength or position of the GS. The EMD analysis also indicates non-stationary variations with
changing amplitude and period over time, showing larger oscillations in all modes after the 1960s,
though this might also be related to a decreasing performance in the sea level reconstruction before the



1940s, when the tide gauge records become much sparser. It is acknowledged that the correlation cannot
indicate exact mechanism or cause-and-effect and each mode may represent a combination of different
mechanisms. For example, for oscillations on time scales of 10-40 years AMO lags behind the GS by 2-
5 years (the $2^{nd}$ and $3^{rd}$ panels in Fig. 4), but for longer time scales (bottom panel of Fig. 4) the GS lags
behind the AMO by 5-10 years. The positive correlation between low frequency variations in the GS
and the AMO can be interpreted in several ways- during periods of more intense flow the GS transports
more heat to the North Atlantic, thus raising SST and increasing the AMO index (i.e., AMO lags behind
the GS), but on the other hand, the AMO is connected to slow variations in AMOC that after some delay
can impact the GS (i.e., GS lags behind AMO).

**231 3.2. Comparison of the reconstructed sea level and the proxy Gulf Stream with recent data**

Very few data sets are long enough to evaluate the entire 116 years of the reconstruction. However,
various recent observations can be used to examine how well the global reconstruction can resolve
regional and basin-wide dynamic processes. The focus here is on three types of observations: coastal sea
level, AMOC and the Florida Current.

**237 3.2.1 Coastal sea level**

The long tide gauge record (starting in 1927) at Sewells Point in Norfolk, VA (in the lower Chesapeake
Bay) has been the subject of many studies due to the acceleration in flooding at this city (Boon, 2012;
Ezer and Corlett, 2012; Ezer, 2013; Ezer and Atkinson, 2014); this location can be used to represent sea
level variability in the MAB (Ezer et al., 2013). Note that due to the course resolution, the reconstruction
completely omits the Chesapeake Bay. The reconstructed sea level also neglects land subsidence, which
is substantial in Norfolk (Boon, 2012; Ezer and Corlett, 2012; Kopp, 2013). Moreover, the altimeter data
that was used in the reconstruction do not extend to the near coast area or to rivers and bays, so that
comparisons between tide gauge data and altimeter data often show that small-scale and high frequency
variations in coastal sea level are not well represented in altimeter data, but interannual and decadal
variations are captured quite well (e.g., see Fig. 2 in Ezer, 2015). Therefore, a comparison of this tide
gauge with the reconstruction (basically a 1°x1° box offshore the Chesapeake Bay) will indicate what
portion of the coastal sea level variability has origin in the offshore large-scale dynamic variability. Fig.





5 shows that while interannual variations in the reconstruction are highly correlated with the tide gauge,
variability in the reconstruction is only about one half of the coastal observations. The correlation of
~0.8 is generally consistent with comparisons made in Dangendorf et al. (2019) for other locations and
may indicate that about 60% of the coastal sea level variability is not locally generated (at least for
monthly data- hourly or daily data may have more influence from local atmospheric forcing and tides).
The reconstruction may not evenly represent all time scales, so to examine this point the variability in
the coastal sea level and in the reconstructed sea level are decomposed into EMD modes (Fig. 6). While
statistically significant correlation (at 95% confidence) is found at all modes, the amplitudes of the
variations are underestimated for high frequency oscillations. In Fig. 7 the EMD modes of the observed
and reconstructed sea level are compared. While the reconstruction captured almost perfectly the mean
frequency of each observed mode (Fig. 7a), the variability of the reconstruction is underestimated by
about a factor of two for the whole time series (mode 1) and for oscillations with periods T<~5 years
(Fig. 7b). For longer time scales (modes 7-10) the reconstruction captured the coastal variability
extremely well with correlations of ~0.9-1. The lowest frequency of oscillating mode 10 in Fig. 6 is
almost identical in the reconstructed and observed sea level, showing an apparent positive acceleration
since the 1960s, in accordance with the global acceleration seen in Dangendorf et al. (2019). Modes 6-8
(with periods of 5-20 years) show especially strong oscillations (Fig. 6 and Fig. 7c). Note that much
longer records are needed to study the oscillations of the lowest frequencies when only a few cycles are
available, though unlike spectral analysis methods, the EMD method is able to detect the potential
existence of very low frequency modes from even incomplete cycles.

### 3.2.2 Atlantic Meridional Overturning Circulation (AMOC)

Continuous observations of AMOC transport at 26.5°N are available since 2004 from the RAPID
program (McCarthy et al., 2012; Srokosz et al., 2012; Baringer et al., 2013; Smeed et al., 2014).
Previous studies found connections between AMOC and sea level difference across the GS as derived
from two tide gauges (Ezer, 2015), so it is interesting to examine if the reconstructed GS shows relation
to the observed AMOC. The RAPID/AMOC transport is the combined contribution from three sources,
Upper Mid-Ocean (UMO) due to density gradients, Ekman (EK) transport due to wind-driven flows, and
Gulf Stream transport as observed by the cable across the Florida Current (FC). These three components
and the total AMOC transport are compared with the proxy GS-MAB record for 2005-2015 (Fig. 8).


Shown are the monthly values and the low frequency EMD modes. The low frequency variations in the
total AMOC transport are significantly correlated (p value <0.05) with the GS proxy (R=0.64) and both
show a weakening trend of ~12% over this decade of comparison (Fig. 8a). However, the GS-MAB
proxy is not significantly influenced by the EK (Fig. 8c; R=0.1) or the FC (Fig. 8d; R=-0.1) components
of AMOC. It does seem though that more than 50% of the variability in the GS-MAB is due to the UMO
(R=0.72). Moreover, the weakening trend in the GS-MAB also seems to be due to the weakening in the
UMO (Fig. 8b). The GS-MAB lags by about a year behind changes in the UMO, a result also obtained
in Ezer (2015). Coherent oscillations with periods of ~2-3 years dominate the low-frequency modes for
GS-MAB, UMO, EK and the total AMOC transport. In summary, it is encouraging that despite the
limitation of using only surface and coastal data in the reconstruction, it can capture the variability of
AMOC including changes in the subsurface density field (i.e., UMO).

### 292    3.2.3 The Florida Current (FC)

Though the RAPID/AMOC transport includes the contribution from the FC, the RAPID record is
relatively short (starting in 2004), compared with the longer observed record of the FC, which started in
1982 (with some gaps). Therefore, the FC transport for 1983-2015 is compared with the reconstructed
GS proxy for the MAB and the SAB (Fig. 9). Note that for this period, the FC shows a weakening trend
of -0.03 Sv/yr (~0.9%/decade), compared with a larger recent weakening (~1.5%/decade) for 2005-2015
(Fig. 8d). While these trends are small and not statistically significant, they do represent a potential
acceleration in the slowdown of the FC if they are real. The correlations of the FC with the GS proxy are
larger in the SAB (R=0.58; Fig. 9b) where the GS is closer to the Florida Straits than in the MAB
(R=0.28; Fig. 9a) where the GS is farther downstream from the observed FC. The lower correlation in
the MAB (though statistically significant at 95%) seems due to a phase lag between the upstream SAB
and the downstream MAB. This incoherence between the GS and coastal sea level on the two sides of
Cape Hatteras (i.e., the SAB versus the MAB) was investigated in several recent studies (Woodworth et
al., 2016; Valle-Levinson et al., 2017; Domingues et al., 2018; Ezer, 2019). EMD analysis further
compares relationship between the GS-SAB proxy (derived from east-west sea level difference) and the
observed FC for different modes (Fig. 10). The high frequency oscillations of the FC and the GS-SAB
are not significantly correlated, in fact, oscillations at ~2-year period show a small but non-significant
anticorrelation at lag zero (second panel in Fig. 10). However, variability on time scales larger than ~5



years are highly correlated (R=0.8-0.9 for modes 6-8 in Fig. 10) with the GS-SAB lagging behind the
observed FC transport; this low frequency variability in modes 6-8 represents cycles with periods of ~5
years, ~12 years and ~24 years, respectively (see right panels in Fig. 10). While theoretically it is
expected that sea level difference across the GS will be correlated with the FC, it is encouraging that a
course global reconstruction with 1 degree resolution that does not resolve the GS front can still capture
the majority of the low frequency variability of the FC. It is noted that although the reconstruction is
based on satellite altimeter data that started in the 1990s, ocean dynamic variability in the 1980s, before
the satellite age, is still captured quite well.

**4. Summary and conclusions**
Since continuous coverage of global sea level from satellite altimeters started only in recent decades
(since the middle 1990s), and century-long tide gauge records are sparse, it is a challenge to study long-
term variations in sea level (decades to multi-decades and longer) with existing data. Such studies are
important for understanding natural variations, anthropogenic changes, and the increased risk to coastal
communities from climate change and sea level rise. To overcome the lack of past data and sparse tide
gauges data, various statistical optimization techniques were used to reconstruct past global sea level.
Here, a new hybrid reconstruction by Dangendorf et al. (2019) for 1900-2015 was examined, with two
main goals in sight: first, to evaluate whether the global coarse resolution reconstruction can capture
regional coastal sea level variability and changes in ocean dynamics and second, to evaluate the
reconstruction against recent observations. The focus of the study was on the southwestern North
Atlantic Ocean, where the dynamics are dominated by the variability of the Gulf Stream system, and
where offshore GS dynamics are an important driver of coastal sea level rise and variability (Blaha,
1984; Leverman et al., 2005; Ezer, 2001, 2013, 2015, 2019; Ezer et al., 2013; Salenger et al., 2012; Yin
et al., 2013; Domingues et al., 2018).
Close examination of the southwestern North Atlantic region in the reconstructed sea level
shows uneven acceleration at different periods during the 116-year record, with larger acceleration in the
last two decades than that of previous periods, as indicated globally in Dangendorf et al. (2019) and
others. However, regionally, the largest changes in sea level rise rates are found near the GS with often
opposing sea level changes north and south of the GS front, thus pointing to the hypothesis that changes
in the GS strength and position may play important roles in the climate variability. To study variations in





the GS, a proxy of the GS strength was derived from sea level differences across the GS in two subregions, the SAB and the MAB. Long-term (time scales longer than 5 years) variations in the reconstructed GS were found to be correlated with the low frequency oscillations of the AMO. Another interesting result is that during the 116-year record, there are two distinct periods of significant weakening in the GS flow, each one lasts for at least a decade when the maximum trend was a declining flow of about 20-25% per decade. The first period with a slowing down GS was seen in the 1960s and 1970s. This period of weakening circulation was previously identified by limited observations (Levitus, 1989, 1990) and early basin-scale diagnostic models (Greatbatch et al., 1991; Ezer et al., 1995) that suggested up to 30% slowdown in the GS transport over a 15-year period (though model results could not be verified due to lack of observations at the time). This weakening was suggested to relate to changes in the subsurface Atlantic Ocean density field and in the wind-driven Ekman transport. Regional acceleration in sea level rise along the U.S. east coast due to the weakening GS was also seen in models and data (Ezer et al., 1995). However, only years later, based on more data, the link between weakening in the GS and AMOC and accelerated coastal sea level became a topic of considerable research (e.g., Levermann et al., 2005; Yin et al., 2010; Sallenger et al., 2012; Ezer et al., 2013). The second period of significant weakening in the reconstructed GS, was the longest in the 116-year record (~1998-2015 and may continue beyond the reconstruction record), though we note that the uncertainties of the reconstruction increases substantially before the 1950s, when tide gauge records become abruptly more sparse. During the more recent period significantly more observations exist that support the recent weakening trend, including altimeter data (Ezer et al., 2013; Ezer, 2015; Dong et al., 2019; Zhang et al., 2020), reconstruction from temperature data (Rahmstorf et al., 2015; Caesar et al., 2018) direct measurements of the GS (Rossby et al., 2014; Andres et al., 2020) and the AMOC/RAPID observations (McCarthy et al., 2012; Srokosz et al., 2012; Baringer et al., 2013; Smeed et al., 2014). A comparison of the reconstructed GS and the observed AMOC shows similar downward trend for 2005-2015 and similar oscillations with periods of 2-5 years. The recent weakening of the reconstructed GS and the variability were correlated with variations in the upper mid-ocean transport component of AMOC and to lesser degree in recent years by changes in the Ekman transport, somewhat resembling processes suggested in the past to explain the 1960s changes.

Another goal of the study was to evaluate the reconstructed sea level against recent observations. Coastal tide gauge data in the lower Chesapeake Bay (in the flood prone city of Norfolk) for 1927-2015 were compared with the reconstructed sea level offshore (the Bay is completely absent from the 1°x1°





coarse resolution reconstruction). Observations of the Florida Current transport for 1982-2015 were also
compared with the reconstructed GS in the SAB. EMD analysis (Huang et al., 1998) was used to
decompose the time series into non-stationary modes of different time-scales in order to examine what
portion of the observed variability can be captured by the reconstruction. The results show that for time
scales of ~5-year and longer, the reconstruction can capture most of the observed variability
(correlations of 0.8-0.9) in both, the coastal sea level and the FC transport.

377          In summary, the study demonstrated that despite the coarse horizontal resolution of the global

reconstruction (1°x1°), and the sparse data that was available before the satellite altimetry age, long-
term variations in regional dynamics can be captured quite well by this global reconstruction, therefore
providing a useful tool for studies of long-term past variability in other regions as well. The long
reconstruction can help studies of decadal and longer natural variability as well as anthropogenic climate
change. For example, the study shows that while the ocean circulation and the GS are subject to natural
multidecadal variations, the recent weakening in the GS is unprecedented in its length during the 116
years of the reconstruction. It also confirmed the existence of another period of significant weakening
GS during the 1960s and 1970s, which previously was suggested only by limited observations. Future
observations are needed to determine if the recent weakening will last due to anthropogenic forces or
recover, like the previous slowdown.

**Acknowledgments:**  The study is part of Old Dominion University's Climate Change and Sea Level
Rise Initiative and the Institute for Coastal Adaptation and Resilience (ICAR). Data used here are
available from the following sites: PSMSL sea level (http://www.psmsl.org/), AMO index
(https://www.esrl.noaa.gov/psd/data/timeseries/AMO/); AMOC transports from the RAPIC project
(http://www.rapid.ac.uk/rapidmoc/)        and        FC        transport        from        NOAA/AOML
(www.aoml.noaa.gov/phod/floridacurrent/). The RecSL data is available by request from the authors.

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





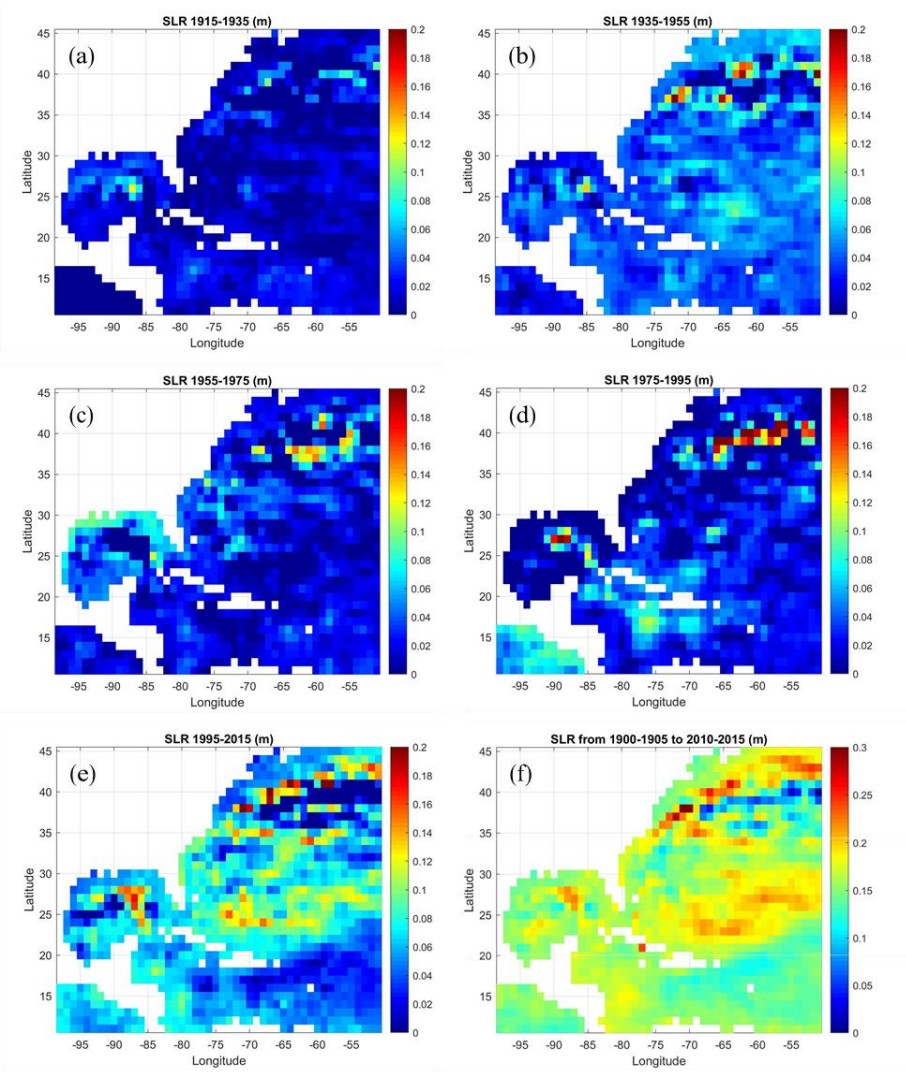

Fig. 1. (a)-(e) Sea level change at different periods. (a) The difference between the mean sea level in 1915 and the mean sea level in 1935, (b) for 1935-1955, (c) for 1955-1975, (d) for 1975-1995, (e) for 1995-2015. Note that the maximum change of 0.2m/20 years is equivalent to a sea level rise of 10 mm/y. (f) Sea level change between the first and last 5 years of the record (note the different color scale).





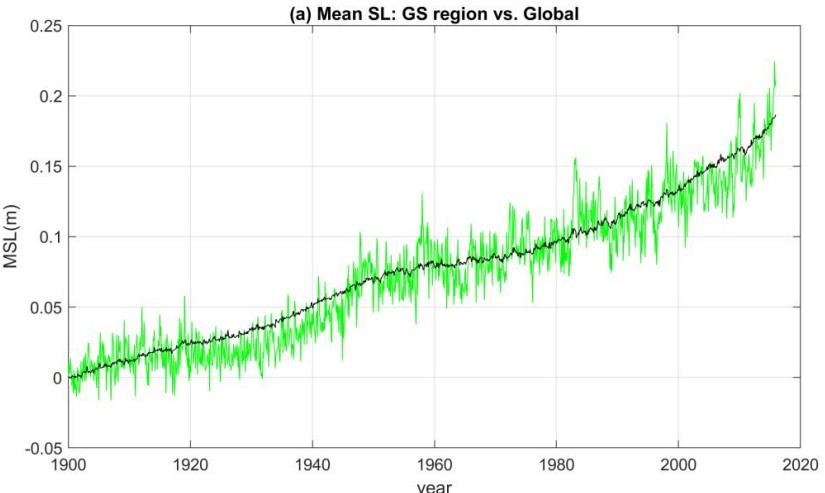

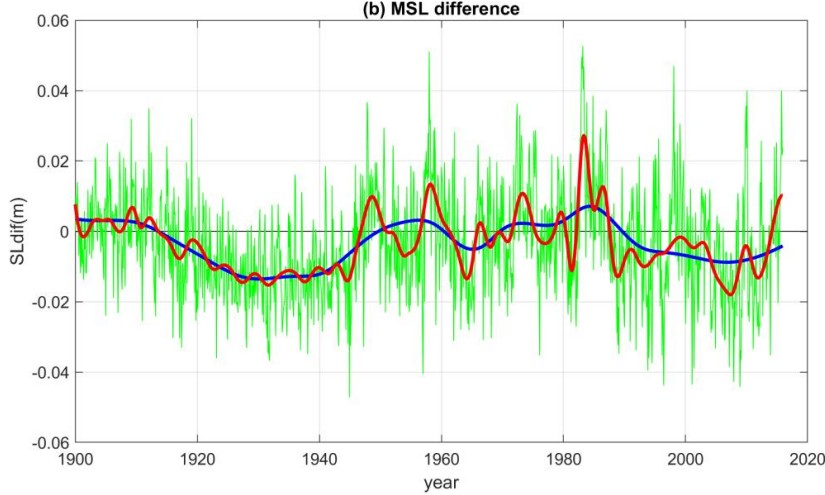

587

Fig. 2. (a) Global mean sea level (black line) and mean sea level over the region shown in Fig. 1 (green

line). (b) Difference between the regional and global mean sea levels (green line). Red and blue heavy

lines represent the low-frequency EMD modes for time scales of ~5-10 years and ~10-60 years,

respectively.

592





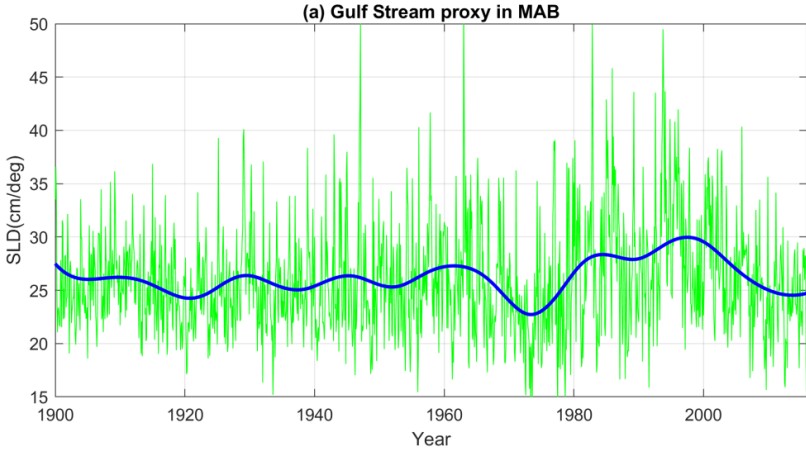

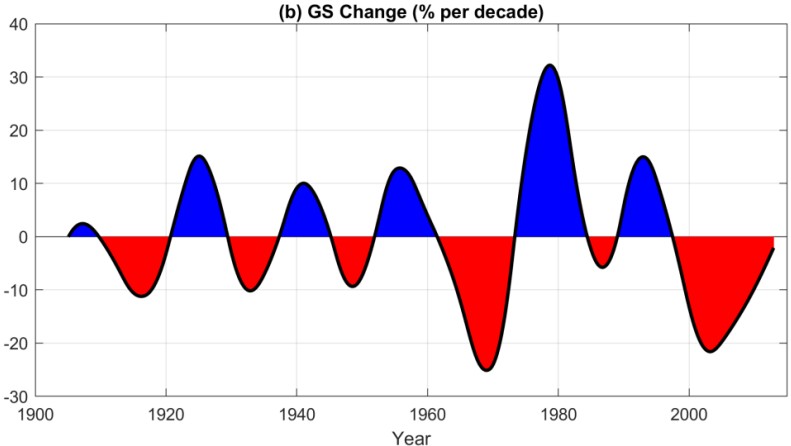

Fig. 3. (a) Gulf Stream (GS) proxy in the Mid-Atlantic Bight (MAB) calculated from the average change
in sea level across the GS in the region (58°W-70°W, 36°N-40°N); the units are cm change per 1°
latitude. Green line is for monthly values and blue heavy line is the low-frequency EMD modes. (b) The
change in the strength of the GS of the low-frequency modes in (a); the units are percentage change per
decade with red/blue represent periods of weakening/strengthening of the GS flow.

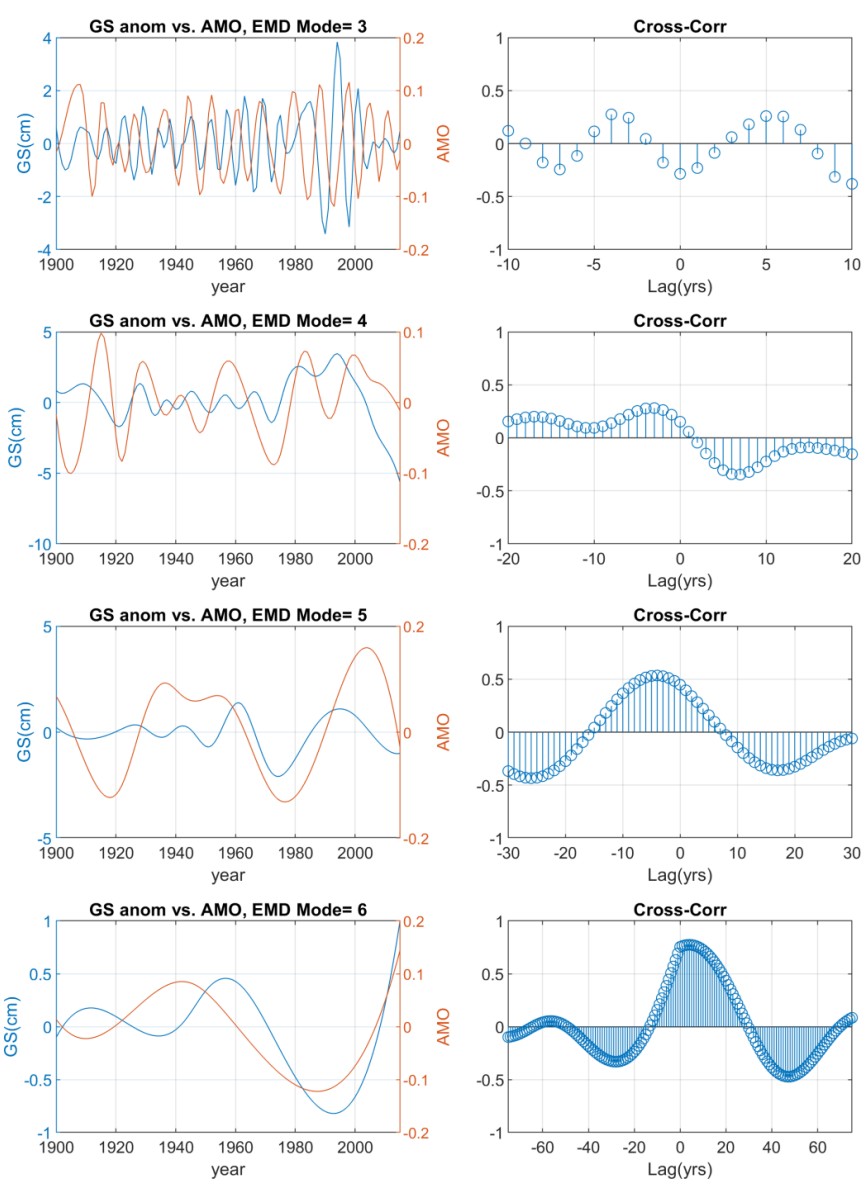


Fig. 4. (a) Comparison of EMD oscillating modes of the monthly GS proxy (blue; units: sea level
change across the GS in cm per degree latitude) and the AMO index (red). (b) Cross correlation as a
function of lag. There are total 7 EMD modes; modes 2-6 are the oscillating modes.



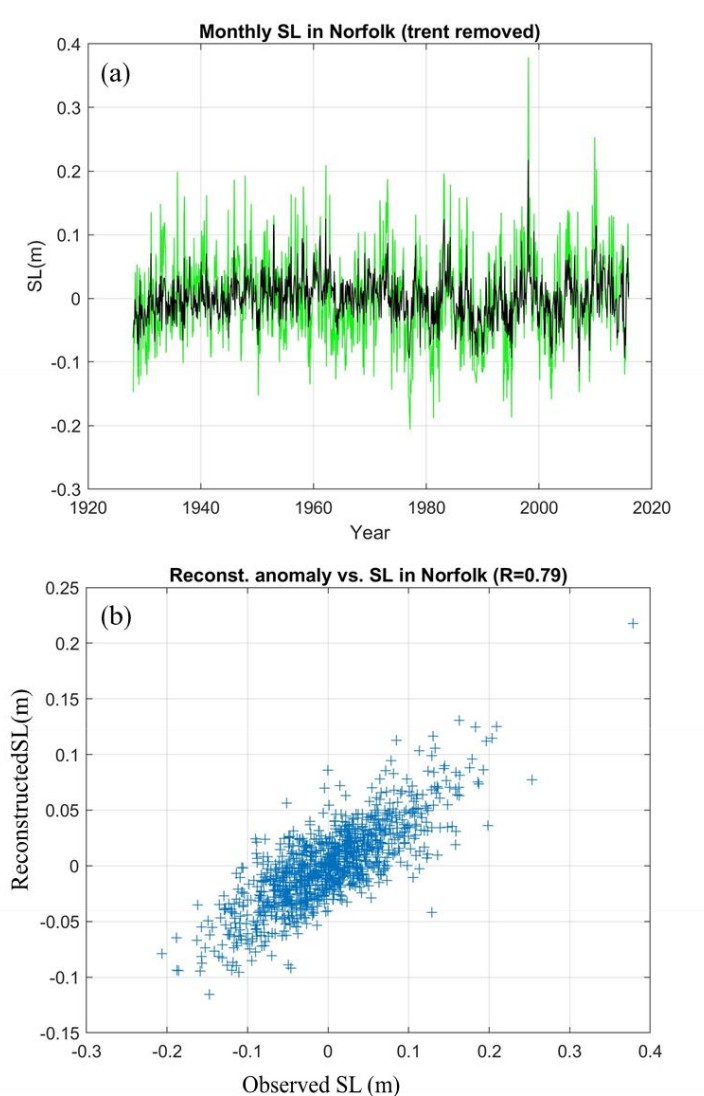


Fig. 5. (a) Comparison of the monthly coastal sea level (green line) observed by the tide gauge near
Norfolk, VA (76.33°W, 36.95°N) and the reconstructed sea level (black line) in the closest 1°x1° box
near the coast. (b) Scatter plot of the data comparison. The trend and the seasonal cycle were removed
from both time series.




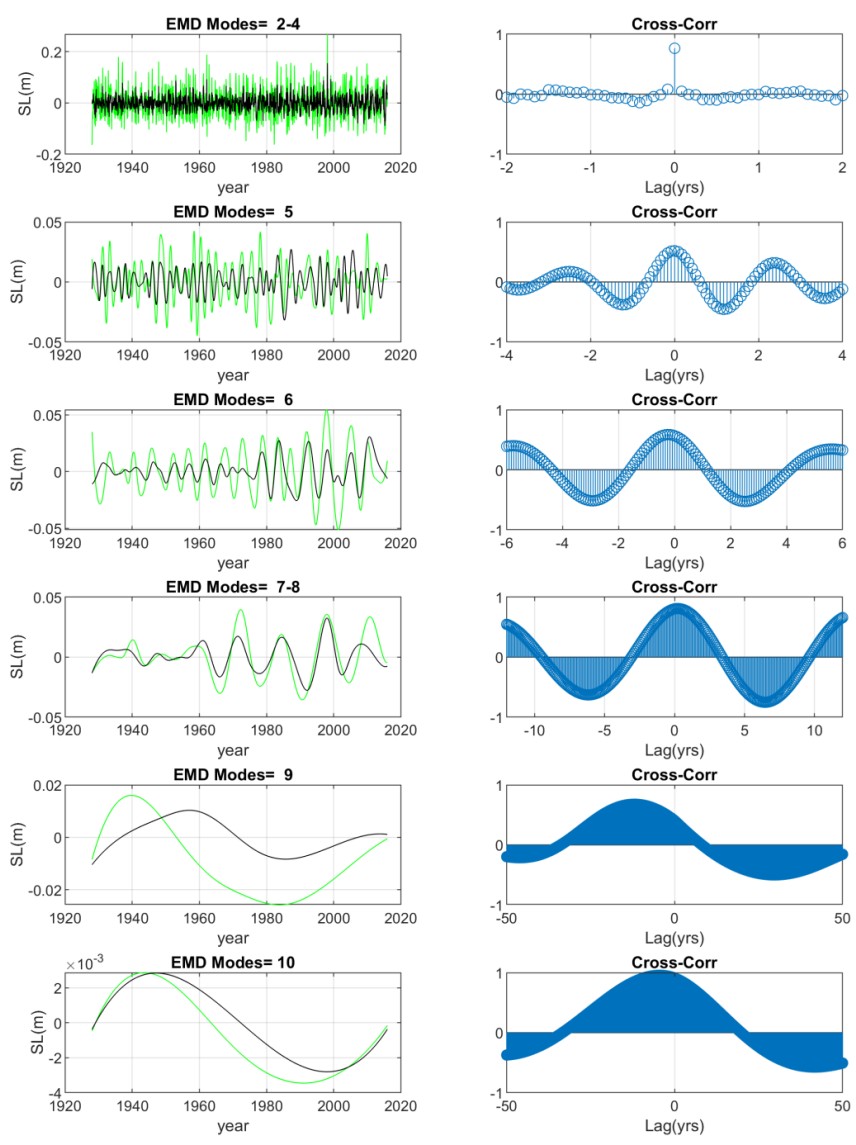


Fig. 6. Left panels: EMD oscillating modes of the Norfolk sea level (green) and the reconstructed sea
level (black). Right panels: Cross-correlation as a function of lag. High to low frequency modes are from
top to bottom panels.





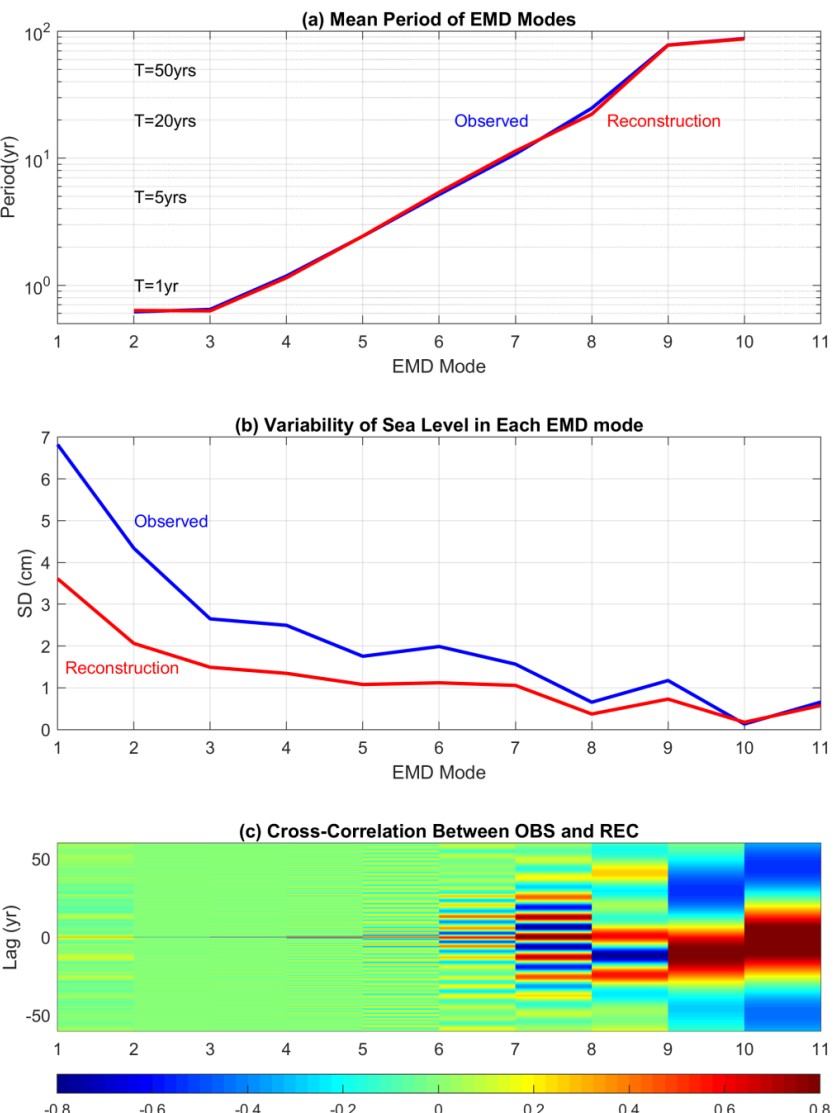


Fig. 7. (a) Mean period of the EMD oscillating modes for the observed sea level (blue) and the

reconstructed sea level (red). (b) Standard deviation of each EMD mode. (c) The cross-correlation

between the observed and reconstructed sea level as function of EMD modes and lag. Note that mode 1

is the original time series, modes 2-10 are oscillating modes (with time-dependent amplitude and

frequency) and mode 11 is the trend.



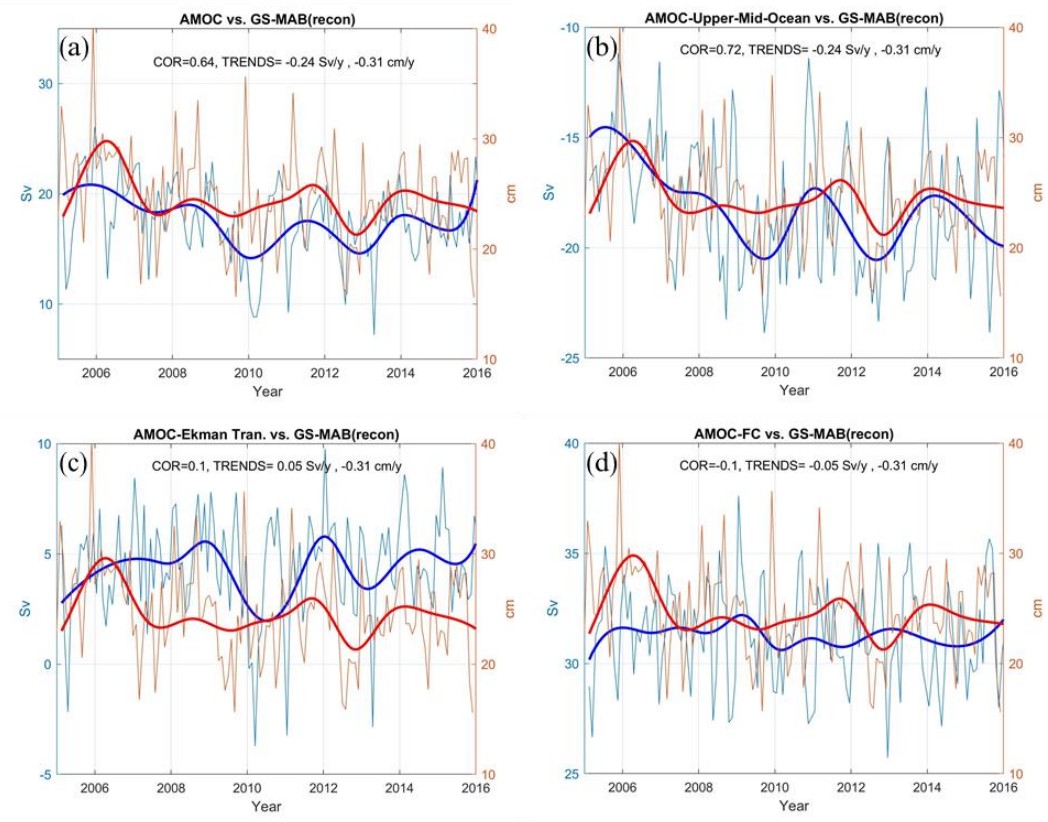


Fig. 8. Comparison between the GS proxy in the MAB (58°W-70°W, 36°N-40°N) and the RAPID
observations: (a) total AMOC transport, (b) upper mid-ocean transport, (c) Ekman transport and (d) the
Florida Current transport. The GS proxy (in blue) is the average north-south sea level change across the
GS (in cm per 1° latitude) representing the eastward flowing strength of the geostrophic surface flow;
RAPID observations (transport in Sv) are in red. Thin lines are monthly values and the heavy lines are
low frequency modes. The correlation of the low frequency modes and the trends of the monthly records
are indicated.




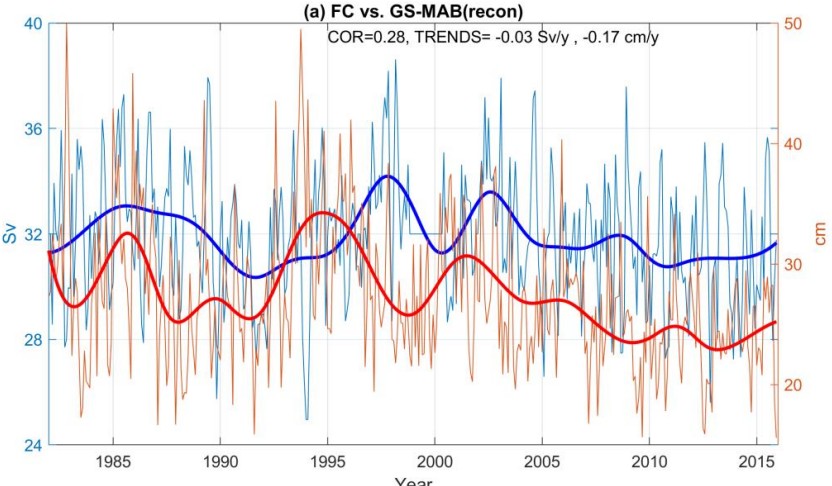

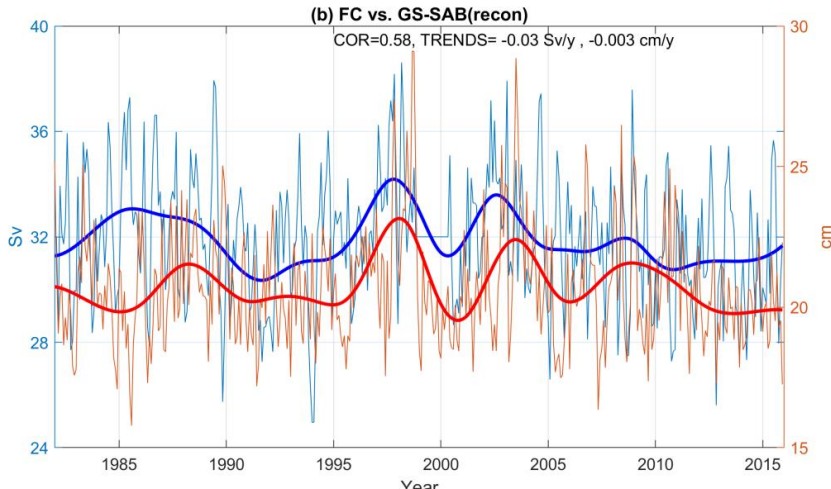


Fig. 9. Comparisons between the observed monthly Florida Current transport (blue, in Sv units on the left) and the GS proxy (red, in cm sea level change across the GS) obtained from the reconstructed sea level difference across the GS for (a) eastward velocity in the MAB (see Fig. 5 for definition) and (b) northward velocity in the SAB (76°W-80°W, 28°N-32°N). Thin lines are monthly values and the heavy lines are low frequency modes. The correlation of the low frequency modes and the trends of the monthly records are indicated.



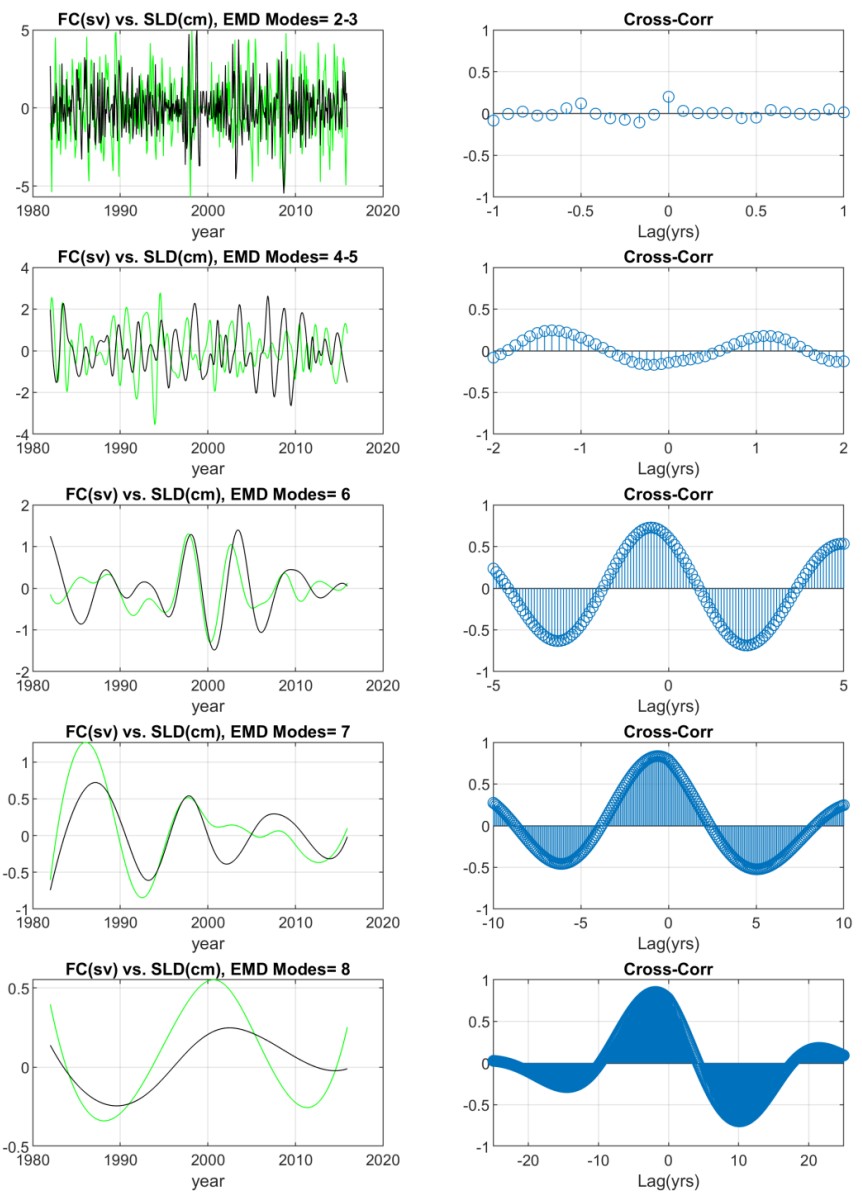


Fig. 10. Left panels: EMD oscillating modes of the observed Florida Current transport (green, in Sv) and
GS proxy in the SAB from the reconstructed sea level (black, in cm). Right panels: Cross-correlation as
a function of lag. High to low frequency modes are from top to bottom panels.