# Peer review of "Global sea level reconstruction for 1900-2015 reveals regional variability in ocean dynamics and an unprecedented long weakening in the Gulf Stream flow since the 1990s"

_Ocean Science, 2020_

## Referee Comment (RC1) · Anonymous Referee #1 · 20 May 2020

Review of Ezer and Dangendorf, "Global sea level reconstruction for 1900-2015 reveals regional variability in ocean dynamics and an unprecedented long weakening in the Gulf Stream flow since the 1990s".

The paper creates two Gulf Stream proxies from a reconstructed sea level dataset (RecSL), compares these against observations (validation), and analyses for trends and relationship to coastal sea level.

[Figure]

I think it's a great idea to analyse this dataset for these purposes. I found some of the validation very compelling and some unfulfilling. I thought the coastal sea level section was under-explored.

I found the structure confusing and suggest it could be rearranged so that validation is separate from the deeper investigation and implications.

This does require major work but, with this, the paper will make an interesting contribution to the literature.

Major comments:

I found the validation of the GS-SAB proxy very compelling (Fig 9b) but much less so the GS-MAB. The GS-MAB proxy as presented does not simply show strengthening/weakening and movement in the position of the GS or broadening of the GS is not considered. For the former, there are long datasets that could have been compared with e.g. Taylor et al. (1998) and Joyce et al. (2000). Better validation of this index and what it represents would make the conclusions more compelling.

The link to coastal sea level could be investigated further. How may your findings be useful for coastal management? I thought the closer correspondence of the index to the coastal sea level at modes of lower variability could be very important. Bingham and Hughes (2009) presented the idea of 1 Sv : 2 cm. How does your reconstruction relate to this? Could this be indicative of differing ocean processes being important in communicating offshore sea level changes to the coast on different timescales?

Structure: validation should precede the implications.

Minor comments:

l77. How would differing modes, not captured in the satellite era, impact the reconstruction?

l89-94, a map illustrating what you have done would be useful here. This would be

beneficial to show the mean SSH from the reconstruction and to compare with the satellite ssh. This would give a better indication to the reader what has been used.

l167-169, you haven't shown us the path of the Gulf Stream. This could simply be a northward shift of the current. There are many papers on the GSNW that discuss this mode of variability. Also, you need to consider whether the GS is weakening or broadening (Dong et al., 2019 —which you cite). The GS could be just as strong but not as narrow.

l196-198, your reference list here is misleading. H&R (2004) showed a strengthening of the AMOC, Dong et al., only spoke of the GS.

This could be a useful indicator but isn't investigated sufficiently accurately.

Conflation of GS and AMOC.

l212-213, what method was used for calculating the degrees of freedom and correlation?

l281, is this your MAB GS proxy?

Fig. 9 b. I find this a very compelling figure.

More updated references for RAPID should be considered: Smeed et al., 2018: https://agupubs.onlinelibrary.wiley.com/doi/full/10.1002/2017GL076350 Moat et al., 2020: https://www.ocean-sci-discuss.net/os-2019-134/

References:

Taylor, A. H., & Stephens, J. A. (1998). The North Atlantic Oscillation and the latitude of the Gulf Stream. Tellus A, 50(1), 134–142.

Joyce, T. M., Deser, C., & Spall, M. A. (2000). The relation between decadal variability of subtropical mode water and the North Atlantic Oscillation. Journal of Climate, 13(14), 2550–2569.

[Figure]

Bingham, Rory J., and Chris W. Hughes. "Signature of the Atlantic meridional overturning circulation in sea level along the east coast of North America." Geophysical Research Letters 36.2 (2009).
* * *

---

## Referee Comment (RC2) · Anonymous Referee #2 · 20 May 2020

The presented article uses the global gridded RecSL sea level reconstruction dataset. It illustrates that this dataset quite accurately represents decadal changes in two time series, the coastal sea level at Sewells Point (Chesapeake Bay) and the Gulf Stream strength for which independent reconstructions exist. This is a result that encourages the application of the dataset in regional long-term studies. I only suggest a few clarifications and corrections of minor flaws, which have no influence on the article's results.

[Figure]

Specific suggestions:

L57-58: Are there other studies that have investigated this question, then please give references, or is this study the first one?

L100: How was the deseasonalisation performed?

L124: How was the gap treated? Did you fill it artificially or did you use analysis methods capabale of dealing with incomplete data?

L133: Please check whether the $c_i$ are actually cumulated sums of modes rather than the independent modes themselves. E.g., the red curve shown in Fig. 2b contains the low-frequency signal of the blue curve, so you probably show cumulated sums over the last (lowest-frequency) independent modes. In this case, $c_1$ will be the full signal, but your formula given in this line would be incorrect. Only the sum over the independent modes, not over their cumulative sums, will give the full signal. Please check.

L163: Sea level cannot accelerate, only sea level rise can. Also, your expression "accelerating over the entire period" is misleading, since within the period, both acceleration and deceleration exist.

L164: Fig. 1f does not show an acceleration. Please define how you calculate acceleration and show a corresponding figure.

L168: In contrast to this sentence, none of your figures indicate a sea level drop anywhere, since their color scales start at zero. Please clarify that.

L179: It is actually the 1970s-1980s only that show higher than average sea level.

L212-213, L257: Please provide more detail on how the significance test was performed, possibly in an online supplement.

L228: Please give some reference to existing literature describing these connections, e.g., Sévellec and Fedorov, Journal of Climate, 2013

L238: I assume the selected tide gauge station or any station in its close proximity did not contribute to the RecSL reconstruction used here, so it can serve as an independent validation site? Please explain this in the text.

Technical corrections:

L44, L45: minima -> minimum

L63: rely -> relies

L64: compared -> compare

L71: summery -> summary

L139: ensemble -> ensembles?

L192: course -> coarse

L193: and impact -> and the impact?

L214: indicated connection -> indicated a connection

L221-222: cannot indicate exact mechanism or cause-and-effect -> please rephrase

L241: course -> coarse

L252: made in Dangendorf et al. -> made by Dangendorf et al.

L314: course -> coarse

L357: increases -> increase

L363: shows similar downward trend -> shows a similar downward trend

Fig. 5: trent -> trend

Fig. 7c: The labels on the x axis should be placed in the middle of the vertical stripes.

---

## Referee Comment (RC3) · Tarmo Soomere (Referee) · 27 May 2020

This is an interesting study of temporal variations in the large-scale circulation and associated water level near the U.S. East Coast. Even though the spatial resolution of the reconstructions is comparably low and single coastal features and even quite large water bodies such as the Chesapeake Bay are not represented at all, the results seem to capture many interesting items described in other studies.

The analysis leads to several interesting points, including the observation that the changes to (the intensity of) Gulf Stream have clear temporal pattern of decrease: one event that happened half century ago and another during this millennium.

The conclusions draw a more dramatic picture than one can observe from the images. For example, lines 383–384 tell that "the recent weakening in the GS is unprecedented in its length during the 116 years of the reconstruction." This is of course true but Figure 3 makes clear that the initial level of the relevant proxy was much higher at the end of the 1990s than in 1960s and the recent weakening more resembles a relaxation of an intense stream back to (or just a little bit below) its usual (almost pre-industrial) level. I would even suggest to adjust the title accordingly.

I suggest to critically look at this and similar claims and to make sure to the reader that the results should not heat up the discussion of accelerating climate change.

Discussion of the acceleration of sea level rise seems to use slightly too much jargon. Acceleration, by definition, is the rate of change of speed. The presence of different rates of the increase in sea level during different time periods does not necessarily mean acceleration over any longer time interval. The description on lines 160–164 only confirms that the average rates are different. The change in rates may happen instantly. Thus, the claim that "there is clear acceleration of sea level over the entire period" is not really substantiated and should be reformulated.

I understand that almost everybody talks/writes today about acceleration of sea level rise but it is better to keep in mind the classic notions of such categories. A larger increase rate (=speed) does not necessarily mean acceleration (=change in speed). The same applies to formulations on line 335 where "larger acceleration" is, most probably, not really a good description of what has happened in the study area.

I agree with the comment of Referee #2 that, in general, the paper is not really well structured. The text should be divided into much shorter paragraphs. Doing so would also make easier to distinguish the results from conjectures.

[Figure]

I thus recommend publication with moderate revisions (mostly better structuring the text and separation of validation from implications and conjectures).

Minor and technical comments (the typos spotted by Referee 1 omitted in the list)

Usually en-dash is used to denote a range of years (e.g. 1900–2015) in Ocean Science

Abstract, line 2 from bottom; also line 55, line 81, line 248, line 370, line 378: use multiplication sign instead of letter "x"

Line 2: apparently should be "show acceleration of global sea level rise" or similar

Lines 16, 17 / line 28 / line 41 and in several occasions below: please unify the use of "U.S. East Coast" / "U.S. East coast" / "U.S. east coast"

Lines 109–111: unify capitalization of "1. The" etc.

Line 128 and in many occasions below: use italics (or better math mode) for mathematical symbols in text like r(t)

Line 298–299: the sentence "While these trends are small and not statistically significant, they do represent a potential acceleration in the slowdown of the FC if they are real", in essence, is an expression of wishful thinking. Either remove or make sure that this kind of conjecture cannot be made based on such very weak changes (in total less than 2%) that basically are at the border of detection.

Line 410: page numbers or article number missing

Line 421: check capitalization

Line 464: remove full stop after "Future"

Line 471: add full stop after "Res"

Line 483 & line 488: volume and page numbers (or article number) are missing

Line 544: page numbers or article number missing

---

## Author Comment (AC1) · 1 Jun 2020

**Response to anonymous Referee #1**

The referee is thanked for the positive comments and many useful suggestions that helped to improve the manuscript. All the comments were addressed as detailed below.

The main changes in the revised manuscript include:

**1.** New Fig. 1 that compares the mean SSH of RecSL with AVISO and shows the locations of data and subregions.

**2.** Reorganization of the entire manuscript, so that validation/evaluation of the reconstruction against data comes first (new section 3.2) and precedes the discussion of mechanisms and basin-scale modes (new section (3.3).

**3.** Seven new references (4 of which were suggested by the reviewer) were added.

Note: because of the reorganization of the paper, track-changes would be too messy, so instead here is the summary of the changes:
Figures: 1 (new fig.), 2 (prev. #1), 3 (#2), 4-6 (#5-7), 7-8 (#9-10), 9 (#3), 10 (#4), 11 (#8)
Sections: 1-2 (same as previous sections)
      3.1. Regional and global sea level rise (previous: Sea level and Gulf Stream)
      3.2. Comparison of the reconstruction with recent data (prev: Comparison & proxy GS)
            3.2.1 Coastal sea level
            3.2.2 The Florida Current
      3.3. Potential driving mechanisms for decadal variability in the RecSL
            3.3.1 The Atlantic Multi-decadal Oscillation (AMO)
            3.3.2 Atlantic Meridional Overturning Circulation (AMOC)
      4. Summary and conclusions (same as previous section)

Response to specific comments:

**4. Ref#1:** *I think it's a great idea to analyse this dataset for these purposes. I found some of the validation very compelling and some unfulfilling. I thought the coastal sea level section was under-explored.*
**Response:** Thanks for the positive comment. We further explore the sea level in a new separate section (3.2.1), but also indicate that extensive validation of SL was already done in Dangendorf et al. (2019), see lines 112-117.

**5. Ref#1:** *I found the structure confusing and suggest it could be rearranged so that validation is interactive separate from the deeper investigation and implications. This does require major work but, with this, the paper will make an interesting contribution to the literature.*
**Response:** This is an excellent idea that makes a lot of sense, so the manuscript was reorganized accordingly (see #2 above).

**6. Ref#1:** *I found the validation of the GS-SAB proxy very compelling (Fig 9b) but much less so the GS-MAB. The GS-MAB proxy as presented does not simply show strengthening/weakening*

*and movement in the position of the GS or broadening of the GS is not considered. For the former, there are long datasets that could have been compared with e.g. Taylor et al. (1998) and Joyce et al. (2000). Better validation of this index and what it represents would make the conclusions more compelling.*

**Response:** These two references were added, and we acknowledge that variations in the position of the GS and in broadening its front may also contribute to spatial changes in SLR. However, as explained now and shown in Fig. 1, the RecSL grid is too coarse (~100 km!) to resolve variations in the GS position or broadening, thus we can only analyze regional means here (lines 192-200).

**7. Ref#1:** *The link to coastal sea level could be investigated further. How may your findings be useful for coastal management? I thought the closer correspondence of the index to the coastal sea level at modes of lower variability could be very important. Bingham and Hughes (2009) presented the idea of 1 Sv : 2 cm. How does your reconstruction relate to this? Could this be indicative of differing ocean processes being important in communicating offshore sea level changes to the coast on different timescales?*

**Response:** The implications are discussed in the conclusions. Also, the Bingham & Hughes reference was added, with the comment that our results suggest somewhat similar relation, around 1Sv : 1.5 cm (lines 180-181, 268-271).

*Structure: validation should precede the implications.* **Done- see #2 & #5 above.**

**8. Ref#1:** *Minor comments:*
*How would differing modes, not captured in the satellite era, impact the reconstruction?*

**Response:** Very good question. It is explained now that multidecadal modes are not resolved by the satellite altimetry era, so low-frequency variations in the hybrid RecSL record are mostly derived from the tide gauge records through the Kalman Smoother, while altimeter data contributing mostly to interannual to decadal variability (lines 77-84, 211-214).

**9. Ref#1:** *l89-94, a map illustrating what you have done would be useful here. This would be beneficial to show the mean SSH from the reconstruction and to compare with the satellite ssh. This would give a better indication to the reader what has been used.*

**Response:** Following this suggestion, a new Fig. 1 compared mean SSH in the RecSL to satellite altimeter data and show the discussed subregions and the location of various observations used.

**10. Ref#1:** *l167-169, you haven't shown us the path of the Gulf Stream. This could simply be a northward shift of the current. There are many papers on the GSNW that discuss Interactive this mode of variability. Also, you need to consider whether the GS is weakening or broadening (Dong et al., 2019). The GS could be just as strong but not as narrow.*

**Response:** The reviewer is correct, but as explain above (point#6), the RecSL data is too coarse to detect shifts in the GS or changes in the width of the front- this kind of analysis had been done in previous cited studied using altimeter or section data which are better fit for that.

**11. Ref#1:** *l196-198, your reference list here is misleading. H&R (2004) showed a strengthening of the AMOC, Dong et al., only spoke of the GS. This could be a useful indicator but isn't investigated sufficiently accurately. Conflation of GS and AMOC.*

**Response:** This paragraph was rewritten (lines 300-313).

**12. Ref#1:** *l212-213, what method was used for calculating the degrees of freedom and correlation?*
**Response:** Detailed information with a new reference to the statistical method used (Thiebaux and Zwiers, 1984).) is now provided to explain how we estimate the degrees of freedom and confidence levels of correlations using EMD modes (lines 158-167).

*Fig. 9 b. I find this a very compelling figure.* **Thanks- this figure is now 7b.**

*More updated references for RAPID should be considered: Smeed et al., 2018:* **Added.**
**Other suggested references that were added:** Taylor & Stephens (1998), Joyce et al. (2000), Bingham et al. (2009).

---

## Author Comment (AC2) · 1 Jun 2020

**Response to anonymous Referee #2**

The referee is thanked for the positive comments and many useful suggestions that helped to improve the manuscript. All the comments were addressed as detailed below.

The main changes in the revised manuscript include:

**1.** New Fig. 1 that compares the mean SSH of RecSL with AVISO and shows the locations of data and subregions.

**2.** Reorganization of the entire manuscript, so that validation/evaluation of the reconstruction against data comes first (new section 3.2) and precedes the discussion of mechanisms and basin-scale modes (new section (3.3).

**3.** Seven new references (1 of which was suggested by this reviewer) were added.

Response to specific comments:

**4. Ref#2:** *The presented article uses the global gridded RecSL sea level reconstruction dataset. It illustrates that this dataset quite accurately represents decadal changes in two time series, the coastal sea level at Sewells Point (Chesapeake Bay) and the Gulf Stream strength for which independent reconstructions exist. This is a result that encourages the application of the dataset in regional long-term studies. I only suggest a few clarifications and corrections of minor flaws, which have no influence on the article's results.*
**Response:** Thank you for the kind words on our study.

**5. Ref#2:** *L57-58: Are there other studies that have investigated this question, then please give references, or is this study the first one?*
**Response:** We added here reference to a recent study by Gehrels et al. (2020), who to our knowledge is the only other study so far that used the Dangendorf et al.'s (2019) RecSL, in this case to study past sea-level rise hotspots along the western North Atlantic Ocean coasts and their relation to the North Atlantic Oscillation (NAO) and to Arctic ice melt.

**6. Ref#2:** *L100: How was the deseasonalisation performed?*
**Response:** This is quite straight forward and explained now (lines 104-108)- the mean annual cycle (averaged over time for each month of the year) was simply calculated and removed.

**7. Ref#2:** *L124: How was the gap treated? Did you fill it artificially or did you use analysis methods capabale of dealing with incomplete data?*
**Response:** It is explained now (lines 132-139) that the EMD analysis can easily handle uneven sampling intervals and data gaps. In fact, the impact of data gaps on EMD analysis of sea level records has been specifically tested in Ezer et al. (2016).

**8. Ref#2:** *L133: Please check whether the $c_i$ are actually cumulated sums of modes rather than the independent modes themselves. E.g., the red curve shown in Fig. 2b contains the low-*

*frequency signal of the blue curve, so you probably show cumulated sums over the last (lowest-frequency) independent modes.*
**Response:** While ci's represent individual EMD modes correctly in Eq. 1, however, we admit that the description of low pass filtered records (e.g., in Figs. 3, 7, 9 & 11) was confusing, as they actually represent the sum of low-frequency modes. This is now clarified in the text and in figure captions.

**9. Ref#2:** L163: *Sea level cannot accelerate, only sea level rise can. Also, your expression "accelerating over the entire period" is misleading, since within the period, both acceleration and deceleration exist. L164: Fig. 1f does not show an acceleration. Please define how you calculate acceleration and show a corresponding figure.*
**Response:** The reviewer is correct, and the text was changed accordingly. The figure in fact does not show acceleration, only SL change, though it is inferred that changes in SLR rates between one period to another means acceleration/deceleration.

**10. Ref#2:** *L168: In contrast to this sentence, none of your figures indicate a sea level drop anywhere, since their color scales start at zero. Please clarify that. L179: It is actually the 1970s-1980s only that show higher than average sea level.*
**Response:** The reviewer is correct, so the text now clarifies that sea level is rising everywhere, but at some locations or at some periods SLR is faster than at other locations or periods.

**11. Ref#2:** *L212-213, L257: Please provide more detail on how the significance test was performed.*
**Response:** Following also a similar question from Referee#1, more detailed information with a new reference to the statistical method used (Thiebaux and Zwiers, 1984) is now provided to explain how we estimate the degrees of freedom and confidence levels of correlations using EMD modes (lines 158-167).

**12. Ref#2:** *L228: Please give some reference to existing literature describing these connections, e.g., Sévellec and Fedorov, Journal of Climate, 2013*
**Response:** This reference and several others were added, as suggested by both referees.

**13. Ref#2:** *L238: I assume the selected tide gauge station or any station in its close proximity did not contribute to the RecSL reconstruction used here, so it can serve as an independent validation site? Please explain this in the text.*
**Response:** This issue is now explained in more details (lines 110-117). It is true that this tide gauge is not completely independent from RecSL, however, the hybrid reconstruction has been validated thoroughly using random independent unassimilated sites (see supplementary material in Dangendorf et al., 2019) and there are so many tide gauge records along the U.S. East Coast that the inclusion/exclusion of a single site such as Sewells Point will have a negligible effect on the reconstructed fields.

**14. Ref#2:** Technical corrections…**Response:** All these typos and text corrections were fixed.

**15. Ref#2:** Fig. 5: trent -> trend, Fig. 7c: The labels on the x axis should be placed in the middle.
**Response:** The 2 figures were redrawn with those corrections (new Figs. 4 & 6).

---

## Author Comment (AC3) · 1 Jun 2020

**Response to Referee #3 (Tarmo Soomere)**

The referee is thanked for his thoughtful comments and suggestions, see our response below.

**1. Ref#3:** This is an interesting study of temporal variations in the large-scale circulation and associated water level near the U.S. East Coast. Even though the spatial resolution of the reconstructions is comparably low and single coastal features and even quite large water bodies such as the Chesapeake Bay are not represented at all, the results seem to capture many interesting items described in other studies. The analysis leads to several interesting points, including the observation that the changes to (the intensity of) Gulf Stream have clear temporal pattern of decrease: one event that happened half century ago and another during this millennium.

**Response:** Thank you for recognizing the important contribution of our study.

**2. Ref#3:** The conclusions draw a more dramatic picture than one can observe from the images. For example, lines 383–384 tell that "the recent weakening in the GS is unprecedented in its length during the 116 years of the reconstruction." This is of course true but Figure 3 makes clear that the initial level of the relevant proxy was much higher at the end of the 1990s than in 1960s and the recent weakening more resembles a relaxation of an intense stream back to (or just a little bit below) its usual (almost pre-industrial) level. I would even suggest to adjust the title accordingly. I suggest to critically look at this and similar claims and to make sure to the reader that the results should not heat up the discussion of accelerating climate change. Discussion of the acceleration of sea level rise seems to use slightly too much jargon. Acceleration, by definition, is the rate of change of speed. The presence of different rates of the increase in sea level during different time periods does not necessarily mean acceleration over any longer time interval.

**Response:** Following comments from Referee#2 discussion of acceleration is toned down and replaced by description of periods of increased or decrease SLR rates (section 3.1). The referee is correct that the recent decline (now Fig. 9) may be part of a decadal cycle and relaxation from a period of strong GS, and this is now clarified in section 3.3. We do not think a change in title is needed.

**3. Ref#3:** I agree with the comment of Referee #2 that, in general, the paper is not really well structured. The text should be divided into much shorter paragraphs. Doing so would also make easier to distinguish the results from conjectures.

**Response:** Following the suggestions by Referees#1&2, we indeed changed the entire organization of the paper and the order of sections and figures to make it more logic and readable.

**3. Ref#3: Minor and technical comments.**

**Response:** All those typos and text corrections have been made. Thanks again for carefully reading our paper.

---

## Short Comment (SC1) · 5 Jun 2020

**Jenny Jardine**

jenjar@noc.ac.uk

Received and published: 5 June 2020

This paper was the subject of a journal club discussion at the National Oceanography Centre. This is not intended as an exhaustive review, but the following points were raised during the discussion and we hope that they will be helpful to the authors.

Write up: Jenny Jardine.

Discussion participants: Jenny Jardine, Jo Williams, Ben Barton, Peter Hogarth and 1 other.

Institution: National Oceanography Centre, Liverpool, UK.

**Description:**

The manuscript uses a coarse (10 x 10) global reconstruction of sea level from 1900-2015 to investigate the long-term regional dynamics of the Western Atlantic Ocean and US East Coast. The two main objectives in this study were to evaluate the model's efficiency in capturing regional sea level variability and ocean dynamics, and to evaluate the sea level reconstruction against recent observations. Main conclusions suggested a weakened Gulf Stream during the late 1960s-70s and since the 1990s that long-term trend analysis using Ensemble Empirical Mode Decomposition (EEMD) correlated to the AMO and AMOC. The manuscript further concluded that the reconstruction was able to adequately capture the regional sea level variability in periods longer than 5 years.

Main Comments:

The paper uses EEMD to separate oscillations at different timescales, yet there was some ambiguity as to what statistical analyses can be done with the output. There is some doubt on the statistical confidence of the EEMD method, given that there are two low frequency oscillations being compared and there is no mention of how many Degrees of Freedom were used to assess the significance of results. As a group, we felt that more description is needed of the EEMD methodology and more confidence that results presented are statistically significant. This could be done by using a comparison to other time-series analysis methods (e.g. EOF analysis), another model output, or more recent datasets in the region (as supplementary material or by referring another paper).

In Figure 2, there is a distinct lack of sea level variability before 1940 that looks a lot
like artificial smoothing. We anticipate this is due to the lack of data during this time period, but this needs to be acknowledged in the main text. Another suggestion is to run the model with random sub-sampled data after 1940 to have a more constant data input. This could show if the increased variability is dependent on the number of data points.

The manuscript also needs a clearer structure. The main conclusion, which reads as the long-term variability in the Gulf Stream, is an interesting result but is quickly lost in the middle of the paper, when the focus shifts to evaluation of regional sea level. One suggestion would be to put Section 3.2 just after the methodology, and then ending with the long-term Gulf Stream variability with AMO/AMOC. At present, it reads like two separate manuscripts pushed into one.

Other Comments:

Line 104: a brief explanation of the ocean dynamics mentioned would be useful. Throughout the manuscript, a more detailed (though still very brief) description of processes would be preferable when listing several references to explain a point

Suggest replacing "weakening trend" to "weakened"

Section 4: this is a very detailed/lengthy section that could benefit from some further summarisation. Much of the information here would be best going into the introduction or discussion sections

Fig 1: the jet colour scheme is slowly being phased out, due to the sharp colour contrasts. Suggest using another colormap

Fig 2: the red-green lines in 2b (and other figures throughout the manuscript) may be difficult for colour-blindness. Suggesting using different colours. Figure 2a would also benefit from an error bar.

OSD

---

## Referee Comment (RC4) · Anonymous Referee #4 · 10 Jun 2020

The 'Global sea level reconstruction for 1900–2015 reveals regional variability in ocean dynamics and an unprecedented long weakening in the Gulf Stream flow since the 1990s' by Ezer and Dangendorf analyzes and discuss the applicability of the global sea level reconstruction for regional studies of ocean dynamics. Despite the relatively coarse resolution of the reconstruction the authors show that low frequency variability is well correlated with coastal sea level and the upper mid ocean transport of the RAPID observations. The authors show that recent transport variations in the Gulf Stream are

represented in the reconstruction and are consistent with findings of other studies.

The claim at the end of the abstract, that the reconstruction is useful for studies of long-term variability in other regions has a good perspective to become true when the reconstruction is analyzed with respect to dynamics and climatic change in other regions. The paper is an important and useful contribution by evaluating the potential of the global sea level reconstruction of Dangendorf et al., 2019. I would recommend publication in Ocean Science after minor revisions.

General comments:

The analysis focuses on the low frequency modes of sea level differences in the Mid-Atlantic Bight but summarizes the findings in terms of periods of sea level rise acceleration. There could be more support for the claim of sea level rise acceleration with a trend analysis instead of a frequency analysis.

Specific comments:

The region defined to calculate the GS-MAB proxy does not include some of the strongest signals shown in the maps in Fig. 1. It would be useful for the reader to have some sort of statement about the sensitivity of the conclusions from Section 3.1 on the definition of the GS-MAB region.

The caption of Fig. 1 suggests that the sea level differences between individual years are shown. The text describing Fig. 1 does not specify details. As the authors point out, the interannual variability (Fig. 2) in the southwestern North Atlantic is high and the maps in Fig. 1 might not be representative for sea level rise in the 20-year periods shown. Maybe a form of average sea level rise during the different 20-year periods would be more appropriate?

I appreciate the placement of the findings into the larger picture in paragraph from l179 to l206. It is on the other hand more like a discussion of the results and the authors might consider to move this part to Section 4.

Fig. 8d and Fig. 9a show a somewhat different correlation between the GS-MAB proxy and the FC transport. It would be interesting to discuss whether this difference is related to the lengths of the time series or to other factors.

l378: long-term variations in regional dynamics can be captured quite well by this global reconstruction, therefore providing a useful tool for studies of long-term past variability in other regions as well.

This statement is probably true, but it has to be shown in the future.

Technical details:

l71: summary

l102: large acceleration (increase?) in flooding

l134: mode-1

l149: acceleration in global mean sea level (rise?) since the

l241: coarse resolution

l261: mode 1

Fig. 5a: trend removed

Fig. 8: sea level change across the GS (in cm per 1 latitude)

Fig. 10: (Sv)

---

## Author Comment (AC4)

**Response to Short Comments from Referee #4 (Jenny Jardin)**

**Ref #4:** *This paper was the subject of a journal club discussion at the National Oceanography Centre. Main conclusions suggested a weakened Gulf Stream during the late 1960s-70s and since the 1990s that long-term trend analysis using Ensemble Empirical Mode Decomposition (EEMD) correlated to the AMO and AMOC. The manuscript further concluded that the reconstruction was able to adequately capture the regional sea level variability in periods longer than 5 years.*
**Response:** The referee is thanked for her minor comments and suggestions, and we appreciate the attention given to this study by UK/NOC's scientists. NOC includes many experts on sea level rise, climate change and scientists monitoring AMOC with the RAPID data used here. I must admit that numerous studies I published using the AMOC data were motivated by and evolved from interactions I had with NOC scientists during a sabbatical I spent in Southampton some 6 years ago, so thanks again, T.E.

**Ref#4:** *There is some doubt on the statistical confidence of the EEMD method, given that there are two low frequency oscillations being compared and there is no mention of how many Degrees of Freedom were used to assess the significance of results. As a group, we felt that more description is needed of the EEMD methodology and more confidence that results presented are statistically significant.*
**Response:** Following similar suggestions from Referees #1 and #2, the revised manuscript provides more details on the EMD method and the statistics used to estimate the degrees of freedom and confidence levels of low frequency EMD modes (lines 158-167). A new reference on the statistics used was also added (Thiebaux and Zwiers, 1984).

**Ref#4:** *In Figure 2, there is a distinct lack of sea level variability before 1940 that looks a lot like artificial smoothing. We anticipate this is due to the lack of data during this time period, but this needs to be acknowledged in the main text. Another suggestion is to run the model with random sub-sampled data after 1940 to have a more constant data input. This could show if the increased variability is dependent on the number of data points.*
**Response:** The referee is correct in the assumption that lack of data at the beginning of the record (no altimeter data and fewer tide gauges) is likely the cause of decrease variability at that time and this is acknowledged in the paper. The original reconstruction paper (Dangendorf et al., 2019) further evaluated the variability, but this is beyond the scope of this paper and would not affect the main results here.

**Ref#4:** *The manuscript also needs a clearer structure. The main conclusion, which reads as the long-term variability in the Gulf Stream, is an interesting result but is quickly lost in the middle of the paper, when the focus shifts to evaluation of regional sea level. One suggestion would be to put Section 3.2 just after the methodology, and then ending with the long-term Gulf Stream variability with AMO/AMOC.*
**Response:** Following a similar suggestion from Referees#1, we indeed changed the entire organization of the paper and the order of sections and figures to make it more logic and readable (very much like what Ref#4 had suggested; see response to Ref#1).

**Ref#4:** *. Line 104: a brief explanation of the ocean dynamics mentioned would be useful.*

*Throughout the manuscript, a more detailed (though still very brief) description of processes would be preferable when listing several references to explain a point. Section 4: this is a very detailed/lengthy section that could benefit from some further summarisation. Much of the information here would be best going into the introduction or discussion sections*

**Response:** The new reorganization and text editing largely took these comments into account.

**Ref#4**: *Fig 1: the jet colour scheme is slowly being phased out, due to the sharp colour contrasts. Suggest using another colormap. Fig 2: the red-green lines in 2b (and other figures throughout the manuscript) may be difficult for colour-blindness. Suggesting using different colours.*

**Response:** Thanks for the suggestions. In fact, the first author himself is a long-time color blind scientist who is well aware of these issues, but in this particular case he found the figure colors clear enough for his eyes…

---

## Author Comment (AC5)

**Response to Anonymous Referee #RC4**

**Ref #RC4:** *The claim at the end of the abstract, that the reconstruction is useful for studies of long-term variability in other regions has a good perspective to become true when the reconstruction is analyzed with respect to dynamics and climatic change in other regions. The paper is an important and useful contribution by evaluating the potential of the global sea level reconstruction of Dangendorf et al., 2019. I would recommend publication in Ocean Science after minor revisions.*
**Response:** The referee is thanked for the positive comments. Indeed, a follow up research now underway is looking at the dynamics of other regions- it was important though to first demonstrate the potential usefulness of the reconstruction at one region, as done here.

**Ref#RC4:** *The region defined to calculate the GS-MAB proxy does not include some of the strongest signals shown in the maps in Fig. 1. It would be useful for the reader to have some sort of statement about the sensitivity of the conclusions from Section 3.1 on the definition of the GS-MAB region.*
**Response:** New Fig. 1 (see below) shows the 2 chosen regions and compares the RecSL with altimeter data. The text explains that these are the regions with SSH gradients that can be used as a proxy for the GS. The change in SSH (now Fig. 2) is aimed at showing changes around the GS regions and the subtropical gyre that connects them. Since the results are area-averaged, they are not very sensitive to slight changes in the chosen regions.

**Ref#RC4:** *The caption of Fig. 1 suggests that the sea level differences between individual years are shown. The text describing Fig. 1 does not specify details. As the authors point out, the interannual variability (Fig. 2) in the southwestern North Atlantic is high and the maps in Fig. 1 might not be representative for sea level rise in the 20-year periods shown. Maybe a form of average sea level rise during the different 20-year periods would be more appropriate?*
**Response:** The caption (new Fig. 2) and associated text were edited and are clearer now. We think that showing the spatial changes (new Fig. 2) and temporal changes (new Fig. 3) are sufficient demonstrations of the differences between global and regional changes, while the focus of the paper should be on the dynamics in the following sections.

**Ref#RC4:** *I appreciate the placement of the findings into the larger picture in paragraph from l179 to l206. It is on the other hand more like a discussion of the results and the authors might consider to move this part to Section 4.*
**Response:** Following suggestions by Referee #RC1, the organization of the paper has been largely changed to make the results and conclusions more logical and clearer.

**Ref#RC4:** *Fig. 8d and Fig. 9a show a somewhat different correlation between the GS-MAB proxy and the FC transport. It would be interesting to discuss whether this difference is related to the lengths of the time series or to other factors.*
**Response:** Thanks for the suggestion, a very good point. Indeed, the FC record used by RAPID (new Fig. 11d) is shorter than the entire FC measurements (new Fig. 7a), thus the differences in modes captured and in correlations- an explanation of this point was added (lines 360-362).

**Ref#RC4:** *l378: long-term variations in regional dynamics can be captured quite well by this global reconstruction, therefore providing a useful tool for studies of long-term past variability in other regions as well. This statement is probably true, but it has to be shown in the future.*
**Response:** As mentioned before, we are studying now the RecSL in other regions, with some promising preliminary results.

**Ref#RC4:** *Technical details:*
**Response:** All those typos and text edits were corrected. Thanks again.

[Figure]

Fig. 1. Mean sea surface height in the North Atlantic Ocean during the satellite era (1993-2015) obtained from (a) the RecSL reconstruction on a 1°×1° grid and (b) the AVISO altimeter data on a 1/4°×1/4° grid. Note the different color scale. The regions where the proxy Gulf Stream is defined in the Mid-Atlantic Bight (MAB) and the South Atlantic Bight (SAB) are marked in (a) and the location of the observations of the Norfolk sea level and the Florida Current transport at the Florida Straits (FLST) are marked in (b).

---

## Author Response (AR1)

**Response to all referees and explanation of the revision.**

Dear Editor,

We thank the referees and others who helped to improve the manuscript, the revision includes all the suggestions and more. All referees had positive comments on the importance of the study and suggested relatively minor revisions and clarifications. Detailed replies to all the comments of the 3 anonymous referees (RC1, RC2, RC4) and two named referees (RC3, SC1) were posted online; we appreciate the great interest the study received from the referees and others (close to 400 views). We also received informal comments (not posted online) from another leading sea level expert, who suggested additional calculations to confirm our results, so additional statistical calculations are added as Supplementary material. The changes in the revised manuscript are detailed in the online replies and highlighted in the "changes" file below.

The main changes in the revised manuscript include:

**1.** New Fig. 1 that compares the mean SSH of RecSL with AVISO and shows the locations of data and subregions.

**2.** Additional calculations were performed, first, to compare the EMD correlations with more traditional wavelet coherence calculations, and second, to test the statistics of the reconstructed GS proxy: 1000 simulations with red noise showed that the found "unprecedented" GS weakening is very unlikely to occur by chance. These calculations are added in 3 new Supplementary figures (S1-S3).

**3.** The structure of the entire manuscript was reorganized, so that validation/evaluation of the reconstruction against data comes first (new section 3.2) and precedes the discussion of mechanisms and basin-scale modes (new section (3.3). The order of sections and all figures was changed accordingly.

**4.** There are 12 additional references: Bingham and Hughes (2009), Chambers (2015), Dangendorf et al. (2017), Ducet et al. (2000), Grinsted et al. (2004), Holgate et al. (2013), Joyce et al. (2000), Montgomery (1938), Sevellec and Federov (2013), Smeed et al. (2018), Taylor and Stephens (1998), Thiebaux Zwiers (1984).

[revised manuscript text omitted]

East Coast relative to the global rates (Boon et al., 2010; Kopp, 2013; Miller et al., 2013; Frederikse et al., 2017; Gehrels et al., 2020). An additional factor, less understood, is acceleration/deceleration due to the dynamic response to changes in ocean circulation, for example, a potential slowdown in the GS and

AMOC (which the GS is part of) can increase coastal sea level along the western North Atlantic coasts (Ezer and Corlett, 2012; Sallenger et al., 2012; Ezer et al., 2013; Ezer and Atkinson, 2014; Rahmstorf et al., 2015; Little et al., 2019). Therefore, it is important to study regional climatic changes for flood- prone coastal communities. The idea of connections between weakening in the GS strength and rising coastal sea level is not new (Montgomery, 1938; Blaha, 1984) and has been identified in data and ocean models (Ezer, 1999, 2001, 2013, 2015; Ezer at al., 2013; Levermann et al., 2005; Yin et al., 2009; Yin and Goddard, 2013; Goddard et al., 2015). Because sea level is lower/higher on the onshore/offshore side of the GS (by ~1-1.5 m; due to the geostrophic balance), changes in the path and strength of the GS

offshore can impact coastal sea level variations along the U.S. East Coast (e.g., see Fig. 2 in Ezer et al.,

2013). This connection involves various temporal and spatial scales and complex mechanisms, so detecting the exact connections between changes in the AMOC and the GS and coastal variability is still an ongoing research (e.g., Little et al., 2019; Piecuch et al., 2019). The processes that transfer large-scale open-ocean signals into coherent regional coastal sea level response involve fast-moving barotropic ocean waves, slow-moving baroclinic waves and coastally trapped waves (Huthnance, 1978; Ezer, 2016;

Hughes et al., 2019). Variations in the GS flow and path have a wide range of time scales: daily, mesoscale, seasonal, interannual, decadal and multidecadal or even longer. However, since direct continuous observations of the GS are relatively short, about 3 decades of satellite altimeter data and about 4 decades of cable observations of the Florida Current (Baringer and Larsen, 2001; Meinen et al.,

2010), it is difficult to study past decadal and multidecadal variability in ocean dynamics and compare it to current and future climate change. For example, limited past temperature and salinity ship observations and simple diagnostic numerical ocean models suggested that a dramatic decline of ~30% in the GS transport happened between the 1960s and 1970s (Levitus, 1989, 1990; Greatbatch et al., 1991); at the same period, an increase in sea level along the U.S. East Coast of 5-10 cm was observed (Ezer et al., 1995). These changes in the 1960s and 1970s resemble recent changes (i.e., coastal sea level rise during periods of GS weakening), but direct observations of the GS and AMOC were not available at the time, to allow comparisons with recent changes. Using ocean models forced by surface observations since the 1960s Blaker et al. (2014) found similarities between the extreme minimum in AMOC in 2009/2010 and a similar minimum in 1969/1970, but this approach has some shortcomings due to models' errors and lack of accurate surface forcing for earlier years.

One approach to overcome the above limitations of studying long term past changes, is to take advantage of the global coverage of recent altimeter data and combine this data with sparse, but long, tide gauges, to obtain global sea level reconstructions. Various optimization and spatial analysis methods were used to produce global reconstructed sea level (Church et al., 2011; Calafat et al., 2014; Hamlington et al., 2014; Hay et al., 2015; Dangendorf et al., 2019). Here, we used the latest hybrid reconstruction of Dangendorf et al. (2019) (see more details in the next section), since it contains both, spatial and temporal variability, as well as long term trends in sea level. Note that this monthly global reconstruction excludes non-climatic land motion, excludes seasonal cycles and is currently available at $1° \times 1°$ resolution for 1900-2015 (future improvements with assimilating higher resolution ocean models, newly digitized tide gauge data and an extended period are planned). Dangendorf et al. (2019) used this reconstruction to study global sea level acceleration and the influence of southern hemisphere winds on sea level, while Gehrels et al. (2020) used it to study past sea-level rise hotspots along the western North Atlantic Ocean coasts and their relation to the North Atlantic Oscillation (NAO) and to Arctic ice melt. The main goal here is to evaluate the usefulness of this reconstruction to study processes of long-term regional ocean dynamics. The western North Atlantic Ocean was chosen as a test case because of the important role that the GS and AMOC play in the basin's dynamics and the fact that the nearby coasts are considered "hotspots" for sea level rise, as described above. Some questions that the study addresses include: 1. Can a coarse resolution reconstruction that does not resolve sharp fronts like that of the GS be able to capture dynamic variations in a western boundary current? 2. How well does the reconstruction, which relies on a relatively short period of altimeter data and sparse tide gauge data, compare with recent independent observations of Atlantic Ocean circulation features such as the AMOC and the Florida Current? 3. What characterizes the long-term variability of sea level and ocean dynamics and how do recent changes such as weakening AMOC compare with past changes? (are recent changes
unprecedented, or more likely natural modes comparable to past changes over the last century?).

The paper is organized as follows: first, the data and the analysis methods are described in
section 2, then in sections 3 the regional and global trends are compared, the reconstruction is evaluated
against observations and decadal variations are studied, finally, in section 4, summary and conclusions
are offered.

**2. Data sources and analysis methods**

The global reconstructed sea level (RecSL) record (1900-2015) analyzed here is described by
Dangendorf et al. (2019). This RecSL is a hybrid reconstruction based on 479 tide gauge records,
satellite altimeter data, and several geophysical ancillary datasets of contributing processes (e.g.
gravitational, rotational, and deformational effects of mass changes known as "fingerprints", ocean
circulation models and GIA), combining the techniques of the Kalman Smoother (Hay et al., 2015),
optimal interpolation and empirical orthogonal functions (Calafat et al., 2014) at different timescales.
The result is a monthly sea level field on a (1°×1°) grid that includes both variability and trends (though
the annual cycle was removed). The aim here is to examine this global data set for its usefulness in
studies of regional ocean dynamics. The western North Atlantic region is characterized by strong
mesoscale variability, an intense western boundary current (the Gulf Stream) and important coastal
impacts from climate change and sea level rise along the U.S. East Coast. Therefore, it is a challenging
task for a coarse resolution reconstruction, which does not resolve mesoscale features, to accurately
represent the regional dynamics. Fig. 1 shows for example, a comparison between the mean sea surface
height (SSH) in the RecSL and the higher resolution (1/4°×1/4°) AVISO satellite altimeter data (Ducet
et al., 2000). While the RecSL captured the main circulation patterns in the North Atlantic Ocean, the
coarse resolution reconstruction is more noisy and underestimate spatial SSH gradients (note however,
that fronts in each monthly field are more defined than in the long-term mean field).

From the reconstructed sea level, a proxy of the GS strength was derived for two regions (Fig.
1a). Based on the assumption that the surface flow is close to geostrophic balance, the sea level gradient
across the GS represents the strength of the surface GS. A shortcoming of this proxy is that it may not
capture subsurface changes. In the Mid-Atlantic Bight (MAB), for each longitude the GS location is
defined by the maximum north-south sea level gradient, so the averaged maximum gradient represents the mean eastward flowing GS in the region (58°W-70°W, 36°N-40°N). The units are change in cm per 1° latitude. In the South-Atlantic Bight (SAB) similar latitudinal averaging of east-west gradients will represent the mean northward flowing GS in the region (76°W-80°W, 28°N-32°N), i.e., between the Florida Strait and Cape Hatteras. These two proxies will be referred to as GS-MAB and GS-SAB, respectively.

The monthly mean sea-level record (1927-2015) for the tide gauge station in Sewells Point near Norfolk (76.33°W, 36.95°N; Fig. 1b) was obtained from the Permanent Service for Mean Sea-level (PSMSL, www.psmsl.org; Woodworth and Player, 2003; Holgate et al., 2013). Since the RecSL record does not include seasonal variability (Dangendorf et al., 2019), the mean annual cycle was calculated and removed from the tide gauge data, to allow a fair comparison. The Norfolk station at the southern end of the Chesapeake Bay was chosen because it is one of the U.S. cities currently facing some of the largest impacts of sea level rise and increased flooding. This tide gauge record was subject to numerous studies that link coastal sea level there with changes in ocean dynamics (Ezer, 2001, 2013; Ezer and Corlett, 2012; Ezer et al., 2013; Ezer and Atkinson, 2014). While this tide gauge was part of the reconstruction, so it is not completely independent from RecSL, the hybrid reconstruction has been validated thoroughly using random independent unassimilated sites (see supplementary material in Dangendorf et al., 2019). There are so many tide gauges along the U.S. East Coast that the inclusion/exclusion of a single site such as Sewells Point will have a negligible effect on the reconstructed fields.

The Atlantic Meridional Overturning Circulation (AMOC) data was obtained from the RAPID observations at 26.5°N for 2005-2015, as described in various studies (https://www.rapid.ac.uk/; McCarthy et al., 2012; Srokosz et al., 2012; Smeed et al., 2014, 2018). The AMOC transport (given in Sverdrup; $1 \text{ Sv} = 10^6 \text{ m}^3 \text{ s}^{-1}$) is the sum of three components: 1. The upper mid-ocean transport obtained from observations of density changes across the Atlantic Ocean, 2. The Ekman transport estimated from wind stress data, and 3. The Gulf Stream transport obtained from cable measurements of the Florida Current across the Florida Strait. These three components of AMOC are provided twice-daily, but they were used here only to calculate monthly averages.

The annual Atlantic Multi-decadal Oscillation (AMO) index (Enfield et al., 2001) for 1900-2015 was obtained from NOAA (https://www.esrl.noaa.gov/psd/data/timeseries/AMO/); AMO represents variations in the sea surface temperature (SST) over the Atlantic Ocean. Long-term variations in sea level, such as a ~60-year long cycle, are thought of being influenced by AMO (Chambers et al., 2012)

and correlations of AMO with patterns of sea level along the U.S. and European coasts are often
indicated (Ezer et al., 2016; Han et al., 2019).

Daily observations of the Florida Current (FC) transport at ~27°N (see Fig. 1b) for 1982-2015
were obtained from NOAA/AOML (www.aoml.noaa.gov/phod/floridacurrent/); the data is described by
Baringer and Larsen (2000), Meinen et al. (2010) and many other studies. Monthly averaged values
were calculated to allow comparisons with the RecSL record. Note that the FC data had a gap from
October 1998 to June 2000. However, the EMD analysis as a sifting/filtering process (described below)
can easily handle uneven sampling intervals and data gaps, so it can detect variations on time scales
longer than the gaps- this has been experimentally tested for long tide gauge records (see Fig. 8 in Ezer
et al., 2016).

A useful tool to analyze non-linear time series is the Empirical Mode Decomposition (EMD
(Huang et al., 1998; Wu et al., 2007), where a repeated sifting process decomposes records into a finite
number of intrinsic oscillatory modes $c_i(t)$ and a residual "trend" $r(t)$. The number of modes depends on
the record length and the variability of the data. Unlike regression fitting methods, the shape of the trend
is not predetermined (i.e., the method is "non-parametric"). Each individual mode does not necessarily
represent a particular physical process, but often a group of modes can be shown to relate to a known
forcing (Ezer et al., 2013; Ezer, 2015). The EMD decomposes the original time series into modes

$$\eta(t) = \sum_{i=2}^{N-1} c_i(t) + r(t).$$ (1)

In the EMD analysis output, mode-1 will be the original time series ($\eta$), modes 2 to N-1 are oscillating
modes with different frequencies from high to low and mode-N will be the trend ($r$). Combining several
low-frequency modes will be equivalent to a low-pass filter. Note that unlike spectral analysis, the
frequency and amplitude in each mode is not constant, thus the analysis can capture non-linear changes,
such as climatic changes in the amplitude of decadal variability. An improved version of the original
EMD, is the Ensemble EMD (EEMD; Wu and Huang, 2009) used here, where ensemble of simulations
with white noise are averaged. Here, 100 ensemble members are used with white noise of 0.1 of the
standard deviation (see Ezer and Corlett, 2012 and Ezer et al., 2016, for sensitivity experiments with
EEMD parameters and error estimations). The EEMD filters out unphysical modes and is more accurate
for detecting real low frequency variability (Kenigson and Han, 2014). All the calculations here use the
EEMD, though for simplicity the text refers to "EMD". Note that the sum of the low frequency modes
plus the trend will be equivalent to a low-pass empirical filter that will have a lower number of degrees of freedom than the original time series. Therefore, when calculating confidence levels on correlations between EMD modes, the "effective sampling size" or "effective number of degrees of freedom" is estimated following the method suggested by Thiebaux and Zwiers (1984). In this method, autocorrelation is used to estimate the typical time scales of low frequency EMD modes and then the confidence level is adjusted accordingly. Empirical testing showed for example that if for the 116-year long monthly RecSL record correlation coefficient of $R=0.08$ provides 95% confidence level ($P\leq0.05$), to obtain the same confidence level for low frequency modes with autocorrelation time scales of 2, 5 and 10 years, will require $R> 0.25$, 0.35 and 0.55, respectively. There have been discussions on the robustness of the EMD in terms of accurately detecting multi-decadal variability and non-linear trends in sea level records (Chambers, 2015). Therefore, to bolster the EMD based correlation analyses between the GS proxy, FC transport and the AMO, we also applied a wavelet coherence analysis (Grinsted et al., 2004), with the results being presented in the Supplementary Material.

**3. Results**

**3.1. Regional and global sea level rise**

Using the same reconstruction (RecSL) analyzed here, Dangendorf et al. (2019) found besides substantial decadal variability a significant and persistent acceleration in global mean sea level rise since the 1960s. They attributed the initiation of this recent acceleration to shifts in Southern Hemispheric wind patterns driving changes in ocean circulation increasing the ocean's heat uptake. In the western North Atlantic, some studies suggest that acceleration in sea level along the eastern coasts of North America may be related to a slowdown of AMOC and the GS (Leverman et al., 2005; Boon, 2012; Ezer and Corlett, 2012; Sallenger et al., 2012; Yin et al., 2013; Caesar et al., 2018). Future projections from climate models consistently indicate a weakening AMOC (Cheng et al., 2013; Reintges et al., 2017), though with divergent associated sea level responses in different models (Little et al., 2019). Therefore, it is important to understand the AMOC-sea level connection and try to detect current and past changes from observations. Bingham and Hughes (2009), for example, suggested that each 1 Sv weakening in AMOC could raise sea level along the North America coast by ~2 cm. To evaluate regional patterns in sea level rise, the sea level change in the southwestern North Atlantic for different periods was calculated (Fig. 2a-e) as well as the sea level change for the entire record 1900-2015 (Fig. 2f). Two findings emerge from this analysis: First, sea level is rising at very different rates during different periods, for example, from 1915 to 1935 (Fig. 2a) sea level rose in the southwestern North Atlantic region by ~0.02-0.04 m (rate of ~1-2 mm/y; similar to the global rate seen in Fig. 2 of Dangendorf et al., 2019), while from 1995 to 2015 (Fig. 2e) sea level in this region rose by ~0.05-0.2 m (rate of 2.5-10 mm/y). Therefore, there is clearly a faster sea level rise since the 1990s compared with previous periods (i.e., sea level rise acceleration), but the sea level rise is spatially very uneven (Fig. 2f). It also seems that due to decadal variability, some periods experienced even decreasing rate of sea level rise (i.e., deceleration), for example, sea level rise from 1955 to 1975 (Fig. 2c) was slower than sea level rise from 1935-1955 (Fig. 2b). Second, the largest changes are seen near the GS around 35°N-40°N with additional changes on the rim of the subtropical gyre (reduction in sea level difference between the center and edge of the subtropical gyre can be interpreted as a sign of weakening circulation). The total sea level change between the first and last 5 years of the RecSL record (Fig. 2f) shows a faster sea level rise north of the GS (red area) and a slower sea level rise south of the GS (blue area), thus indicating a potential weakening trend in the geostrophic surface flow of the GS- this prospect is investigated later. Variations in the NAO and AMOC can cause changes in the GS position and/or in broadening its front (Taylor and Stephens, 1998; Joyce et al., 2000; Smeed et al., 2018), which can also result in spatial variations in sea level rise as seen here. However, the 1°×1° RecSL grid will not resolve most of the variability in the GS position (Fig. 1a), which nevertheless can be seen by the higher resolution altimeter data (Fig. 1b; see also Fig. 1 in Ezer et al., 2013).

A comparison of the global monthly mean sea level with the regional mean sea level in the southwestern North Atlantic (the area shown in Fig. 2) indicates a similar general trend (Fig. 3a), but a much larger interannual and decadal regional variability of up to ±4 cm over the global mean sea level (Fig. 3b). Regionally lower than average sea level is seen in the 1920s-1940s and higher than average sea level is seen in the 1950s and 1980s. Low-passed filtered data (Fig. 3b) show variations on two major time-scales, periods of ~5-10 years (the sum of EMD modes with periods longer than ~5 years is shown in red) and ~10-60 years (the sum of EMD modes with periods longer than ~10 years is shown in blue). The decadal and multidecadal variations in the global acceleration/deceleration of sea level were described by Dangendorf et al. (2019) and others, but we further want to evaluate here if regional variations in ocean dynamics may play a role and how these variations are connected to basin-scale climate modes (Han et al., 2019). Note that multidecadal modes are not resolved by the satellite altimetry era, so low-frequency variations in the hybrid RecSL record are estimated by the Kalman

Smoother applied on the tide gauge records while altimeter data contributing mostly to interannual to decadal variability (Dangendorf et al., 2019). We will return later to discuss the potential mechanisms behind the regional variability seen in Fig. 3b, but before that it is important to validate the RecSL record and evaluate its ability to capture observed the variability.

**3.2. Comparison of the reconstruction with recent data**

Very few data sets are long enough to evaluate the entire 116 years of the reconstruction. However, various recent observations can be used to examine how well the global reconstruction can resolve regional and basin-wide dynamic processes. The focus here is on two types of observations: coastal sea level and the Florida Current.

**3.2.1 Coastal sea level**

The long tide gauge record (starting in 1927) at Sewells Point in Norfolk, VA (in the lower Chesapeake Bay) has been the subject of many studies due to the acceleration in flooding at this city (Boon, 2012; Ezer and Corlett, 2012; Ezer, 2013; Ezer and Atkinson, 2014); this location can be used to represent sea level variability in the MAB (Ezer et al., 2013). Note that due to the coarse resolution, the reconstruction completely omits the Chesapeake Bay. The reconstructed sea level also removes land subsidence, which is substantial in Norfolk (Boon, 2012; Ezer and Corlett, 2012; Kopp, 2013). Moreover, the altimeter data that was used in the reconstruction do not extend to the near coast area or to rivers and bays, so that comparisons between tide gauge data and altimeter data often show that small-scale and high frequency variations in coastal sea level are not well represented in altimeter data, but interannual and decadal variations are captured quite well (e.g., see Fig. 2 in Ezer, 2015). Therefore, a comparison of this tide gauge with the reconstruction (basically a 1°×1° box offshore the Chesapeake Bay) will indicate what portion of the coastal sea level variability has origin in the offshore large-scale dynamic variability. Fig. 4 shows that while interannual variations in the reconstruction are highly correlated with the tide gauge, variability in the reconstruction is only about one half of the coastal observations. The correlation of ~0.8 is generally consistent with comparisons made by Dangendorf et al. (2019) for other locations and may indicate that about 60% of the coastal sea level variability is not locally generated within the Bay area (at least for monthly data- hourly or daily data may have more influence from local atmospheric forcing and tides). The reconstruction may not evenly represent all time scales, so to examine this point the variability in the coastal sea level and in the reconstructed sea level are decomposed into EMD

modes (Fig. 5). Cross-correlations help to identify the main oscillations in each mode. While statistically significant correlation (at 95% confidence) is found at all modes (see section 2 for details on confidence levels of EMD modes), the amplitudes of the variations are underestimated for high frequency oscillations. In Fig. 6 the EMD modes of the observed and reconstructed sea level are compared. While the reconstruction captured almost perfectly the mean frequency of each observed mode (Fig. 6a), the variability of the reconstruction is underestimated by about a factor of two for the whole time series (mode 1) and for oscillations with periods T<~5 years (Fig. 6b). The underestimation is likely due to the variability in the satellite altimeter data and not due to the reconstruction itself. For longer time scales (modes 7-10) the reconstruction captured the coastal variability extremely well with correlations of ~0.9-1. The lowest frequency of oscillating mode 10 in Fig. 5 is almost identical in the reconstructed and observed sea level, showing an apparent positive acceleration in sea level rise since the 1960s, in accordance with the global acceleration seen in Dangendorf et al. (2019). Modes 6-8 (with periods of 5-20 years) show especially strong oscillations (Fig. 5 and Fig. 6c). Note that much longer records are needed to study the oscillations of the lowest frequencies when only a few cycles are available, though unlike spectral analysis methods, the EMD method is able to detect the potential existence of very low frequency modes from even incomplete cycles.

**3.2.2 The Florida Current (FC)**

 In Fig. 7 the observed FC transport for 1983-2015 is compared with the reconstructed GS proxy for the MAB and the SAB. Note that for this period, the FC shows a small weakening trend of -0.03 Sv/yr (~0.9%/decade), while a larger recent weakening (~1.5%/decade) is seen during the RAPID/AMOC observations of 2005-2015 (see discussion in next section). The correlations of the FC with the GS proxy are larger in the SAB ($R$=0.58; Fig. 7b) where the GS is closer to the Florida Straits than in the MAB ($R$=0.28; Fig. 7a) where the GS is farther downstream from the observed FC (see Fig. 1). The lower correlation in the MAB (though statistically significant at 95%) seems due to a phase lag between the upstream SAB and the downstream MAB. This incoherence between the GS and coastal sea level on the two sides of Cape Hatteras (i.e., the SAB versus the MAB) was investigated in several recent studies (Woodworth et al., 2016; Valle-Levinson et al., 2017; Domingues et al., 2018; Ezer, 2019). It is interesting to note that the relation between low frequency changes in the FC transport and sea level as seen in Fig. 7b implies a ratio of about 1 Sv to 1.5 cm while Bingham and Hughes (2009) suggested a ratio of ~1 Sv to 2 cm between AMOC transport and coastal sea level.

EMD analysis further compares relationship between the GS-SAB proxy (derived from east-west
sea level difference) and the observed FC for different modes (Fig. 8). The high frequency oscillations
of the FC and the GS-SAB are not significantly correlated, in fact, oscillations at ~2-year period show a
small but non-significant anticorrelation at lag zero (second panel in Fig. 8). However, variability on
time scales larger than ~5 years are highly correlated ($R$=0.8-0.9 for modes 6-8 in Fig. 8) with the GS-
SAB lagging behind the observed FC transport; this low frequency variability in modes 6-8 represents
cycles with periods of ~5 years, ~12 years and ~24 years, respectively (see right panels in Fig. 8). This
result is further supported by a complementary wavelet coherence analysis (Supplementary Fig. S1).
While theoretically it is expected that sea level difference across the GS will be correlated with the FC,
it is encouraging that a coarse resolution global reconstruction on a 1-degree grid that does not resolve
the GS front very well can still capture the majority of the low frequency variability of the FC. It is
noted that although the reconstruction is based on satellite altimeter data that started in 1993, ocean
dynamic variability in the 1980s, before the satellite age, is still captured quite well.

**3.3. Potential driving mechanisms for decadal variability in the RecSL**

Variability in the GS-MAB proxy (obtained from sea level gradients as described in section 2) is shown
in Fig. 9a, indicating large variability on interannual and decadal time scales with a persistent weakening
trend since ~1990, after a period of strengthening flow from the 1970s to the 1990s. The changes in the
low-frequency oscillations are shown in Fig. 9b, indicating two long periods with declining GS strength
(red area) during the 1960s and 1970s and after ~1995, with maximum weakening of ~25% per decade.
Recent observations by Andres et al. (2020) at 68.5°W found the GS transport to be about 10% weaker
today than it was in the 1980s at the same location, but the same study also found very large discrepancy
in the trend between two sections located just a few 100 km from each other, a western section from ship
crossing showed no statistically significant trend (Rossby et al., 2014) and an eastern section from
mooring data showed potential weakening of ~5-10% per decade. Based on altimeter data, Dong et al.
(2019) and Zhang et al. (2020) also showed different trends between the eastern and western parts of the
GS. Therefore, average GS proxy over a large area as done here may filter out spatial variations; the
RecSL record is also much longer than the altimeter data used in the above studies. The coarse
resolution of the reconstruction also served as a filter that smoothed out small spatial variations and the
impact from local recirculation gyres as seen in Andres et al. (2020). The GS-MAB proxy here shows
that the recent weakening period is the longest in this record. To test if the long period of GS weakening is "unprecedented" or distinct from random natural variability, statistical analysis with 1000 simulations using random red noise (following an autoregressive process of the order 1) imitating the spectrum of the record in Fig. 9a was performed and the results are shown in Supplementary Fig. S2. This analysis shows that obtaining long periods of weakening from random variability is extremely rare: in 116,000 years of artificial simulations there were only 3 cases of weakening of 10%/decade that lasted for 10 years. For comparison, in the 116 years of reconstructed GS there were 2 such cases, with 10%/decade weakening of ~10 years in the 1970s and ~15 years in the 2000s.

Various mechanisms can affect variations in the GS flow such as changes in the strength of the subpolar gyre circulation (Hakkinen and Rhines, 2004) or weakening in the AMOC (Bryden, 2005; McCarthy et al., 2012; Srokosz et al., 2012; Ezer et al., 2013; Smeed et al., 2014, 2018; Blaker et al., 2014; Roberts et al., 2014; Ezer, 2015; Rahmstorf et al., 2015; Caesar et al., 2018). The earlier period of GS weakening in the 1960s-1970s is consistent with observations and models that showed large density changes in the North Atlantic and as much as 30% weakening in the GS between 1955-1959 and 1970-1974 (Levitus, 1989, 1990; Greatbatch et al., 1991; Ezer et al., 1995). At the time of these early studies, before the age of satellite altimeters, observations were limited and models less sophisticated, so there were some doubts that the large weakening in the GS during the 1960s and 1970s was real. However, this reconstruction by Dangendorf et al. (2019) and another reconstruction of AMOC from sea level data by Ezer (2015) both confirm the results of the early studies, showing only two periods of pronounced weakening AMOC since the 1950s. Future observations will show if the recent decline is just a relaxation from the strong GS of the 1980s and 1990s or a continuous downward trend. The relation between the GS-MAB proxy and basin-scale processes are thus analyzed below by looking at two measures, AMO and AMOC.

**3.3.1 The Atlantic Multi-decadal Oscillation (AMO)**

The large decadal and multidecadal variations in the GS-MAB proxy as seen in Fig. 9 are compared with the annual Atlantic Multi-decadal Oscillation index (AMO; Enfield et al., 2001) for 1900-2015 (Fig. 10). EMD is used to compare oscillating modes with similar time scales. Hi-frequency modes of the GS and AMO are not significantly correlated, but variability in the two time series on time scales of ~10-60 years are correlated, especially the lowest frequency modes (bottom two panels in Fig. 10), with correlations of 0.5-0.8 that are statistically significant at 95% (after considering the reduction in degrees of freedom in the low-frequency modes, see explanation in section 2). The wavelet coherence analysis (Supplementary Fig. S3) principally confirms the sign of these correlations in a 16-year frequency band, though they are not statistically significant. This indicates that the correlations identified by the EMD should be taken as preliminary, requiring further data and analyses that are beyond the scope of this paper. Mode 6 (bottom panel in Fig. 10) indicates cyclic behavior at periods up to ~60 years, consistent with previous studies (Chambers et al., 2012). Various studies indicated a connection between AMO, which represents variations in SST, and sea level. Ezer et al. (2016) for example, showed a change in the sign of the correlation across the GS, which could indicate changes in the GS strength; if sea level rises at one side of the GS and drops at the other side, the change in gradient indicates a change in strength or position of the GS. The EMD analysis also indicates non-stationary variations with changing amplitude and period over time, showing larger oscillations in all modes after the 1960s, though this might also be related to a decreasing performance in the sea level reconstruction before the 1940s, when the tide gauge records become much sparser. It is acknowledged that correlation does not indicate cause-and-effect and that each EMD mode may not necessarily represent a specific mechanism. For example, for oscillations on time scales of 10-40 years AMO lags behind the GS by 2-5 years (the $2^{nd}$ and $3^{rd}$ panels in Fig. 10), but for longer time scales (bottom panel of Fig. 10) the GS lags behind the AMO by 5-10 years. It is interesting to note that modes 4 and 5 captured the minimum GS-MAB in the 1970s, while mode 6 captured the minimum in the 2000s. The positive correlation between low frequency variations in the GS and the AMO can be interpreted in several ways- during periods of more intense flow the GS transports more heat to the North Atlantic, thus raising SST and increasing the AMO index (i.e., AMO lags behind the GS), but on the other hand, the AMO is connected to slow variations in AMOC that after some delay can impact the GS (i.e., GS lags behind AMO). The low frequency multidecadal modes seen here resemble findings from ocean models such as the decadal variations seen in the early Atlantic model of Ezer (1999) and in a realistic Atlantic Ocean model of Sevellec and Federov (2013), who found an oscillatory AMOC mode with a period of ~24 years and an e-folding decay time scale of ~40 years that relates to westward propagation of large-scale temperature anomalies (thus connecting AMOC with the AMO). Therefore, the relation of GS-MAB and the observed AMOC is analyzed next.

**3.3.2 The Atlantic Meridional Overturning Circulation (AMOC)**

Continuous observations of AMOC transport at 26.5°N are available since 2004 from the RAPID program (McCarthy et al., 2012; Srokosz et al., 2012; Baringer et al., 2013; Smeed et al., 2014, 2018).

Previous studies found connections between AMOC and sea level difference across the GS as derived from two tide gauges (Ezer, 2015), so it is interesting to examine if the reconstructed GS shows relation to the observed AMOC. The RAPID/AMOC transport is the combined contribution from three sources,

Upper Mid-Ocean (UMO) due to density gradients, wind-driven Ekman (EK) transport and Gulf Stream transport as observed by the cable across the Florida Current (FC). These three components and the total

AMOC transport are compared with the proxy GS-MAB record for 2005-2015 (Fig. 11). Shown are the monthly values and the low frequency EMD modes. The low frequency variations in the total AMOC

transport are significantly correlated ($P<0.05$) with the GS-MAB proxy ($R=0.64$) and both show a weakening trend of ~12% over this decade of comparison (Fig. 11a). However, the GS-MAB proxy is not significantly influenced by the EK (Fig. 11c; $R=0.1$) or the FC (Fig. 11d; $R=-0.1$) components of

AMOC. Note that the FC record used by RAPID in Fig. 11d is much shorter than the entire FC record in

Fig. 7, thus the longer record captures lower frequency modes and shows higher correlation with the GS-

MAB (Fig. 7a; $R=0.28$). It does seem though that more than 50% of the variability in the GS-MAB is due to the UMO ($R=0.72$). Moreover, the weakening trend in the GS-MAB also seems to be due to the weakening in the UMO (Fig. 11b). The GS-MAB lags by about a year behind changes in the UMO, a result also obtained in Ezer (2015). Coherent oscillations with periods of ~2-3 years dominate the low- frequency modes for GS-MAB, UMO, EK and the total AMOC transport. In summary, it is encouraging that despite the limitation of using only surface and coastal data in the reconstruction, it can capture the variability of AMOC including changes in the subsurface density field (i.e., UMO). It is also noted that in the earlier period of weakening GS in the 1970s (Fig. 9), changes in the Atlantic density field rather than changes in the wind fields were suggested as the main cause of this weakening (Levitus, 1989,

1990; Greatbatch et al., 1991; Ezer et al., 1995), which is consistent with the finding here of the main cause of the recent period of GS weakening.

**428   4. Summary and conclusions**

[revised manuscript text omitted]

Fig. 1. Mean sea surface height in the North Atlantic Ocean during the satellite era (1993-2015) obtained from (a) the RecSL reconstruction on a 1°×1° grid and (b) the AVISO altimeter data on a 1/4°×1/4° grid. Note the different color scale. The regions where the proxy Gulf Stream is defined in the Mid-Atlantic Bight (MAB) and the South Atlantic Bight (SAB) are marked in (a) and the location of the observations of the Norfolk sea level and the Florida Current transport at the Florida Straits (FLST) are marked in (b).

[Figure]

Fig. 2. (a)-(e) Sea level change at different periods. (a) The difference between the mean sea level in

1915 and the mean sea level in 1935, (b) for 1935-1955, (c) for 1955-1975, (d) for 1975-1995, (e) for

1995-2015. Note that the maximum sea level change in the colorbar, 0.2m/20 years, is equivalent to a sea level rise of 10 mm/y. (f) Sea level change between the first and last 5 years of the record (note the different color scale).

[Figure]

[Figure]

Fig. 3. (a) Global mean sea level (black line) and regional mean sea level over the area shown in Fig. 2 (green line). (b) Difference between the monthly regional and global mean sea levels (green line). Heavy red and blue lines represent low pass filtered records obtained from the sum of EMD modes with time scales longer than ~5 years and ~10 years, respectively.

[Figure]

[Figure]

Fig. 4. (a) Comparison of the monthly observed coastal sea level (green line) at the tide gauge near
Norfolk, VA (76.33°W, 36.95°N; see Fig. 1) and the reconstructed sea level (black line) in the closest
1°×1° box near the coast. (b) Scatter plot of the data comparison. The trend and the seasonal cycle were
removed from both time series.

[Figure]

Fig. 5. Left panels: EMD oscillating modes of the observed Norfolk sea level (green) and the reconstructed sea level (black). Right panels: Cross-correlation as a function of lag. High to low frequency modes are from top to bottom panels.

[Figure]

Fig. 6. (a) Mean period of the EMD oscillating modes for the observed sea level (blue) and the reconstructed sea level (red). (b) Standard deviation of each EMD mode. (c) The cross-correlation between the observed and reconstructed sea level as function of EMD modes and lag. Note that mode 1

is the original time series, modes 2-10 are oscillating modes (with time-dependent amplitude and frequency) and mode 11 is the trend.

[Figure]

[Figure]

Fig. 7. Comparisons between the observed monthly Florida Current transport (blue, in Sv units on the left) and the GS proxy (red, in cm sea level change units on the right) obtained from the reconstructed sea level difference across the GS for (a) eastward velocity in the GS-MAB and (b) northward velocity in the GS-SAB. Thin lines are monthly values and the heavy lines are low-pass filtered records (sum of low frequency EMD modes). The correlation of the low frequency modes and the trends of the monthly records are indicated.

[Figure]

Fig. 8. Left panels: EMD oscillating modes of the observed Florida Current transport (green, in Sv) and

GS-SAB proxy from the reconstructed sea level (black, in cm). Right panels: Cross-correlation as a function of lag. High to low frequency modes are from top to bottom panels.

[Figure]

[Figure]

Fig. 9. (a) Gulf Stream proxy in the Mid-Atlantic Bight (GS-MAB) calculated from the average change in sea level across the GS; the units are cm change per 1° latitude. Green line is for monthly values and blue heavy line is the sum of low-frequency EMD modes. (b) The change in the strength of the GS of the low-frequency modes in (a); the units are percentage change per decade with red/blue represent periods of weakening/strengthening of the GS flow.

[Figure]

Fig. 10. (a) Comparison of EMD oscillating modes of the annual GS-MAB proxy (blue; units: sea level change across the GS in cm per degree latitude) and the annual AMO index (red). (b) Cross correlation as a function of lag. There are total 7 EMD modes; modes 2-6 are the oscillating modes.

[Figure]

Fig. 11. Comparison between the GS-MAB proxy and the RAPID observations: (a) total AMOC

transport, (b) upper mid-ocean transport, (c) Ekman transport and (d) the Florida Current transport. The

GS proxy (in blue) is the average north-south sea level change across the GS (in cm per 1° latitude)

representing the eastward flowing strength of the geostrophic surface flow; RAPID observations (transport in Sv) are in red. Thin lines are monthly values and the heavy lines are sum of low frequency

EMD modes. The correlation of the low frequency modes and the trends of the monthly records are indicated.

[Figure]

Fig. S1. Squared wavelet coherence between the standardized FC and GS-SAB time series. The 5% significance level against red noise is shown as a thick contour. All significant sections above 4 years show in-phase behavior, which is indicated by arrows that are directed to the right.

[Figure]

[Figure]

Fig. S2. Histogram of the statistics of the GS-MAB flow changes using 1000 realizations of 116 years with random red noise (total of 116,000 years); the simulations imitate the spectrum of the reconstructed GS in Fig. 9a. In each of these simulations GS change was calculated using EMD as in Fig. 9b. (a) The probability of obtaining different GS changes shows that the chance of GS weakening by over 20%/decade (as seen in the reconstruction) is less than 1% for any particular month. (b) The distribution of period length with GS flow declining by at least 10%/decade shows that there were only 3 cases of GS weakening that last at least 10 years during the 116,000 years. For comparison, the GS-MAB in Fig. 9b shows 2 such cases in 116 years, ~10 year weakening period in the 1970s and ~15 year weakening period in the 2000s.

[Figure]

Fig. S3. Squared wavelet coherence between the standardized AMO and GS-MAB time series. The 5%

significance level against red noise is shown as a thick contour. All significant sections below 10 years show anti- phase behavior, which is indicated by arrows that are directed to the left. Positive, through statistically non- significant, correlations are found in the 16-year bands since the 1960s and confirm the low-frequency modes identified by the EMD in the main paper.